**EMBO** *reports*

# Developmental epigenetic programming by Tet1/3 determines peripheral CD8 T cell fate

Kara M Misel-Wuchter [1,2,3], Andrew L Thurman [3], Jordan T Johnson [4], Athmane Teghanemt [1,3], Neelam Gautam[1,3], Alejandro A Pezzulo [3], Jennifer R Bermick[1,4,5], Noah S Butler[4,6,7] & Priya D Issuree [1,2,3,4 ✉]

## Abstract

In response to infections, naive CD8 T cells give rise to effector and memory T cells. However, eliciting long-lived memory CD8 T cells remains a challenge for many infections. DNA demethylation of cytosines within CpG dinucleotides by Tet enzymes is a key epigenetic mechanism that regulates short- and long-term transcriptional programs in cells. Currently, their roles in modulating CD8 T-cell effector and memory differentiation are unclear. Here, we report that developing CD8 T cells lacking Tet1/3 preferentially differentiate into short-lived effector and effector memory cells following acute infection. Using genome-wide analyses, mice in which Tet1/3 were ablated during T-cell development and mature CD8 T cells, respectively, we show that Tet1/3 regulates these cell fates by licensing the chromatin landscape of genes downstream of T-cell receptor activation during thymic T-cell maturation. However, in mature CD8 T cells, Tet1/3 are dispensable for effector and memory cell fates. These findings unveil context-specific roles of DNA demethylation, which are essential for defining pathways that contribute to CD8 memory T-cell generation in response to infections.

**Keywords** DNA Demethylation; Memory CD8 T Cells; Effector CD8 T Cells; T-Cell Development; Epigenetics
**Subject Categories** Chromatin, Transcription & Genomics; Immunology; Microbiology, Virology & Host Pathogen Interaction

## Introduction

CD8 T cells are uniquely equipped to kill pathogen-infected cells during an infection and are undeterred by point mutations that alter the efficacy of neutralizing antibodies against pathogens. However, eliciting a pool of long-lived functional memory CD8 T cells remains a challenge for many infections (Harty et al, 2000;

Sallusto et al, 2010; van de Wall et al, 2021). Understanding the factors and mechanisms that govern the differentiation of distinct memory subsets is the first step in successfully manipulating the immune system for effective and long-term vaccination outcomes.

During an acute infection, a significant portion of proliferating CD8 T cells undergo terminal differentiation into effector cells, commonly known as short-lived effector cells (SLECs). These cells play a crucial role in eliminating the acute infection and typically succumb to apoptosis, although a surviving subset of these effector-like cells can persist into the memory pool with some memory-like features (Milner et al, 2020; Renkema et al, 2020). In contrast, a subset of early-activated cells generates memory precursor effector cells (MPECs), which exhibit multipotency and can differentiate into various types of long-lived memory cells, including central memory ($T_{CM}$) and effector memory ($T_{EM}$) cells (Chung et al, 2021). These distinct memory cell populations possess distinct homing properties and functional capabilities when responding to secondary infections (Gerlach et al, 2013; Masopust et al, 2001; Wherry et al, 2003).

A large body of evidence suggests that multiple signals, including T-cell receptor (TCR), signaling, inflammation, and metabolic signaling, can orchestrate CD8 T-cell fate decisions early during an immune response (Chung et al, 2021). TCR signaling is deemed a key determinant of memory commitment, function, and the diversity of the memory pool (Chin et al, 2022; Daniels and Teixeiro, 2015; Solouki et al, 2020). The integration of TCR signals to alter cell fate outcomes ultimately depends on both the activation of a transcription factor network and concurrent changes in epigenetic modifications that create a chromatin environment permissive for transcription. How epigenetic processes allow for the propagation of upstream TCR signals and/or enable these signals to be fine-tuned to control cell fate outcomes is an area of ongoing investigation.

A key epigenetic change accompanying the differentiation of CD8 T cells involves alterations in DNA methylation at the fifth carbon of cytosines (known as 5-methylcytosine or 5mC), a modification commonly linked to the suppression of transcription (Cedar et al, 2022; Greenberg and Bourc'his, 2019; Jones, 2012). During acute LCMV infection, effector CD8 T cells experience loss

[1]Inflammation Program, University of Iowa, Iowa City, IA, USA. [2]Molecular Medicine Graduate Program, University of Iowa, Iowa City, IA, USA. [3]Department of Internal Medicine, University of Iowa, Iowa City, IA, USA. [4]Immunology Graduate Program, University of Iowa, Iowa City, IA, USA. [5]Department of Pediatrics, University of Iowa, Iowa City, IA, USA. [6]Department of Microbiology and Immunology, University of Iowa, Iowa City, IA, USA. [7]Graduate Program in Molecular Physiology and Biophysics, University of Iowa, Iowa City, IA, USA. ✉E-mail: priya-issuree@uiowa.edu

of DNA methylation in promoter and enhancer-like regions in genes associated with effector CD8 T-cell function, while cells that give rise to memory cells acquire de novo DNA methylation (Scharer et al, 2013; Youngblood et al, 2017). The de novo methyltransferase Dnmt3a, establishes new methylation marks in genes associated with naive T-cell states such as *Tcf7*, *Cd62l*, and *Ccr7* and is critical for restraining the number of memory precursor effector cells that are generated early during T-cell activation (Ladle et al, 2016; Youngblood et al, 2017).

Meanwhile, the role of DNA demethylation in specifying distinct cell fates during viral infection is less clear. Active DNA demethylation is mediated by the family of Ten-eleven translocation (Tet) methylcytosine dioxygenases, which is comprised of three genes (Tet1, Tet2, and Tet3) that are expressed in T cells (Issuree et al, 2018; Tsagaratou et al, 2017; Wu and Zhang, 2017). To our knowledge, only Tet2 has been examined in the context of memory CD8 T-cell differentiation. Using mice in which Tet2 was conditionally deleted in developing T cells, it was previously shown that loss of Tet2 leads to increased generation of MPEC and $T_{CM}$ CD8 T cells following acute LCMV infection (Carty et al, 2018). The similarity of these phenotypes with Dnmt3a deficiency is striking, given that Dnmt3A catalyzes the de novo methylation of cytosines and Tet2 oxidizes 5mC to 5hmC. The contributions of Tet1 and Tet3 in regulating epigenetic programs during CD8 T-cell differentiation have not been examined.

To shed insights into the role of DNA demethylation on CD8 T-cell outcomes during a viral infection, in this study, we sought to elucidate the roles of Tet1 and Tet3 on CD8 T-cell fate outcomes during acute LCMV infection. We discovered that Tet1/3 were required to restrain the formation of SLECs and favored the differentiation of $T_{CM}$ cells upon acute LCMV infection in a cell-intrinsic manner. Unexpectedly, however, we discovered that modulation of these cell fate programs was not dependent on the activities of Tet1/3 during T-cell priming. By employing genome-wide epigenetic profiling and by utilizing two distinct lines of conditional mice, in which Tet1/3 were ablated during T-cell development and in mature CD8 T cells, respectively, we established the necessity of Tet1/3 during T-cell development. Specifically, Tet1/3 were essential during development to license the chromatin landscape of genes downstream of TCR activation. Tet1/3 supported chromatin accessibility at TCR-responsive regulatory elements enriched in motifs belonging to bZIP, ETS, RUNT, and NFAT transcription factor families, which are known transcriptional regulators for early effector CD8 T-cell differentiation. However, despite reduced chromatin accessibility in Tet1/3-deficient cells, binding of the AP-1 family member JunB, a critical bZIP transcription factor, was intact, suggesting chromatin remodeling is likely required for partnership with other TFs. In contrast, we found that early changes in DNA demethylation following TCR activation did not occur within lineage-defining genes, and Tet1/3 were dispensable for effector and memory CD8 cell fate outcomes in peripheral CD8 T cells. Together, our studies underscore the critical role of Tet1/3 in the early and long-term epigenetic programming of CD8 T cells in the thymus. Most notably, they bring into focus the importance of understanding the unique contributions of Tet enzymes in a context-dependent manner to effectively manipulate epigenetic pathways for optimal vaccination outcomes.

# Results

## Loss of Tet1 and Tet3 in developing CD8 T cells results in increased SLECs and effector memory T-cell differentiation during acute LCMV infection

Infection with the lymphocytic choriomeningitis virus (LCMV) Armstrong strain in C57BL/6 mice leads to an acute infection that is usually resolved within 8 days. At this juncture, the cellular numbers of antigen-specific CD8 T cells have peaked, and the cells have differentiated into effector cells (Wherry et al, 2003). Using Tet2$^{fl/fl}$CD4Cre$^{Tg}$ mice in which developing CD4 and CD8 T cells lack Tet2, it was previously shown that the loss of Tet2 promotes MPEC formation and $T_{CM}$ CD8 T-cell differentiation in a cell-intrinsic manner following acute LCMV-Armstrong infection (Carty et al, 2018). To test the role of Tet1/3 in CD8 effector T-cell fates during acute LCMV-Armstrong infection, we challenged RorcCre$^{Tg}$ Tet1/3$^{fl/fl}$ mice (hereafter referred to as Tet1/3$^{cDKO}$), in which double-positive thymocytes and their αβ CD4$^+$ and CD8$^+$ single positive progenies are deficient in Tet1 and Tet3 (Eberl and Littman, 2004; Issuree et al, 2018). Of note, Tet1/3$^{cDKO}$ mice have comparable numbers of peripheral CD8 T cells compared to controls and quiescent CD4 and CD8 T-cell compartments (Fig. EV1A,B). Furthermore, the loss of Tet1/3 in CD8 T cells did not result in a compensatory increase in Tet2 mRNA expression (Fig. EV1C). Eight days post-infection (dpi) with LCMV-Armstrong, the proportions, and numbers of antigen-experienced Tet1/3$^{cDKO}$ CD8 T cells, as assessed by low CD8a and positive CD11a expression, were modestly increased compared to WT littermate controls (Fig. 1A). The proportions and absolute numbers of SLECs among these cells, denoted by high KLRG1 and negative CD127 expression (Joshi et al, 2007), were significantly higher in Tet1/3$^{cDKO}$ mice (Fig. 1B). Using MHC-I restricted epitope-specific D$^b$-GP33-41, D$^b$-GP276-286 and D$^b$-NP396-404 tetramers, we found that the frequencies of tetramer-positive CD8 T cells in Tet1/3$^{cDKO}$ mice were reduced, although the numerical numbers of these cells were comparable to WT controls (Figs. 1C and EV1D), plausibly due to a higher expansion of non-antigen-specific CD8 T cells in Tet1/3$^{cDKO}$ mice. However, a significantly higher proportion of these epitope-specific T cells displayed a SLEC phenotype compared to WT controls (Figs. 1D and EV1E), suggesting that the propensity of Tet1/3$^{cDKO}$ CD8 T cells to adopt a SLEC cell fate was not restricted to TCR specificities. Despite the increased frequency of SLECs in Tet1/3$^{cDKO}$ mice, the viral copies of LCMV-Armstrong 5 dpi were not significantly different compared to WT mice (Fig. EV1F). $T_{EM}$ and $T_{CM}$ differentiation occur early during a primary T-cell response and persist in secondary lymphoid organs long after the infection is cleared (Chung et al, 2021). These populations can be identified through their differential expression of CD27 and CX3CR1, markers linked to functional maturation and T-cell trafficking in memory cells (Gerlach et al, 2016; Martin and Badovinac, 2018). While both GP33-41-specific $T_{EM}$ and $T_{CM}$ cells were numerically reduced in Tet1/3$^{cDKO}$ mice due to the overall decrease in GP33-41-specific cells at 24 dpi (Figs. 1E and EV1G), we observed a markedly higher proportion of Db-GP33-41-specific $T_{EM}$ cells compared to $T_{CM}$ cells at both 24 and 8 dpi in Tet1/3$^{cDKO}$ mice compared to WT controls (Figs. 1E and EV1G,H). Together, these data suggest that

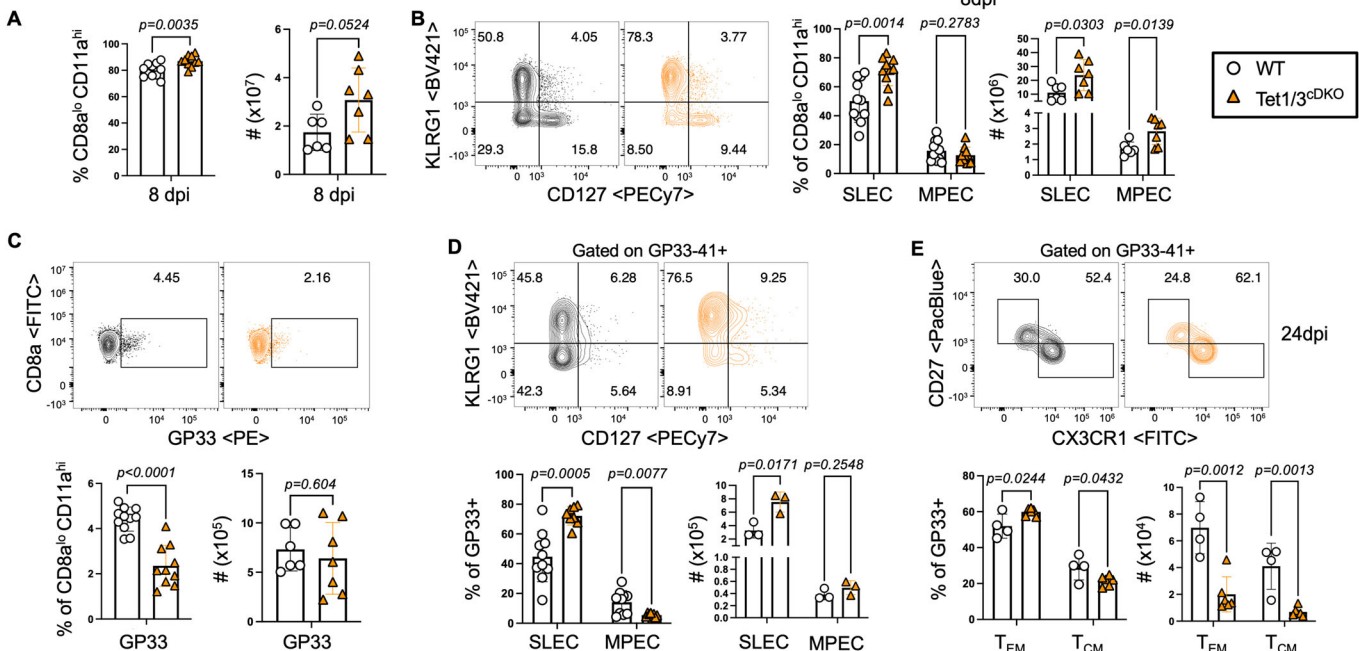

**Figure 1. Tet1 and Tet3 deficiency results in increased SLECs and effector memory T-cell differentiation during acute LCMV infection.**

WT and Tet1/3^cDKO mice were infected with 2×10^5 PFU of LCMV-Armstrong, and (A–D) blood was assessed 8dpi and (E) splenocytes were assessed 24dpi. (A) Percentage and number of antigen-experienced CD8 T cells. (WT = 11, Tet1/3^cDKO = 10 biological replicates) Data are mean ± SEM, unpaired t test. (B) Representative FACS plots comparing SLEC and MPEC populations by KLRG1 and CD127 expression and quantification of frequency and number of SLECs/MPECs, pregated on LiveTCRb+CD8a^lo CD11a^hi cells. (WT = 11, Tet1/3^cDKO = 10 biological replicates) Data are mean ± SEM, multiple unpaired t tests. (C) Representative FACS plots comparing GP33-tetramer+ CD8 T cells and quantification of frequency and number of tetramer-specific CD8 T cells, pregated on LiveTCRβ+CD8a^lo CD11a^hi cells. (WT = 11, Tet1/3^cDKO = 10 biological replicates) Data are mean ± SEM, unpaired t test. (D) Representative FACS plots and quantification of frequency and number of SLECs/MPECs among GP33-tetramer+ cells 8 dpi. (WT = 10, Tet1/3^cDKO = 8 biological replicates) Data are mean ± SEM, multiple unpaired t tests. (E) Representative FACS plots and quantification of frequency and number of T_EM/T_CM cells among GP33-tetramer+ CD8 T cells 24dpi. (WT = 4, Tet1/3^cDKO = 6 biological replicates) Data are mean ± SEM, multiple unpaired t tests. Data information: P values are indicated in the figures and P < 0.05 was considered significant; paired male or female mice were used and no gender biases associated with genotypes were observed. (A–E) Experiments were replicated at least three times. Source data are available online for this figure.

in contrast to the loss of Tet2, loss of Tet1/3 in developing T cells enhances SLECs and T_EM cell formation during acute LCMV infection, highlighting distinct roles for Tet1/2/3 during CD8 T-cell differentiation.

## Tet1/3 restrain the differentiation of SLECs and effector memory T cells in a cell-intrinsic manner

To investigate whether the susceptibility of CD8 T cells to differentiate into SLECs and T_EM cells was due to cell-intrinsic requirements of Tet1/3, we transferred equal numbers of congenically- disparate WT or Tet1/3^cDKO P14 CD8+ T cells, which have a TCR restricted to the MHC-I-specific LCMV glycoprotein (GP) 33-41 peptide (Pircher et al, 1990), into WT hosts and infected them with LCMV-Armstrong a day later (Fig. 2A). LCMV infection led to comparable activation of WT or Tet1/3^cDKO P14 CD8+ T cells (Fig. EV2A), equivalent Ki67 expression in expanding antigen-experienced T cells (Fig. EV2B), and similar recovery of transferred WT and Tet1/3^cDKO T cells 5, 8, 10 and 14 dpi (Fig. EV2C), suggesting that Tet1/3 deficiency does not impact the activation or early proliferation of GP33-41-specific T cells. However, similar to polyclonal CD8 T cells from infected Tet1/3^cDKO mice, a higher frequency of Tet1/3^cDKO P14 cells differentiated into SLECs compared to WT P14 cells at

multiple time points analyzed (Figs. 2B and EV2D). Similarly, Tet1/3^cDKO P14 cells preferentially differentiated into T_EM cells compared to WT controls at day8 and day27 dpi (Figs. EV2E and 2C). While the proportions of activated cells on day27 were similar, the total number of Tet1/3^cDKO P14 cells recovered was reduced (Fig. 2D), suggesting that the retention of this memory pool may be affected, although further longitudinal experiments are required. To ensure that memory populations detected by CD27 and CX3CR1 were bonafide T_EM and T_CM cells, we also assessed the expression of the transcription factor Tbet, which is expressed at higher levels in T_EM cells compared to T_CM cells (Joshi et al, 2007). Consistent with their transcriptional identities, we observed higher expression of Tbet in T_EM cells from both WT and Tet1/3^cDKO P14 T cells compared to the T_CM counterparts, and notably, Tbet expression was not affected by Tet1/3 deficiency (Fig. EV2F). Interestingly, upon ex vivo restimulation of splenocytes (isolated 10 dpi) with GP33-41 peptide, a lower proportion of Tet1/3^cDKO T cells produced IFNγ and TNF-α compared to WT controls (Fig. 2E,F), despite the increased frequency of SLECs and T_EM-like cells among Tet1/3^cDKO T cells, suggesting effector differences in these populations. Taken together, these results suggest that Tet1/3 restrain the differentiation of SLECs and T_EM cells during acute LCMV infection in a cell-intrinsic manner.

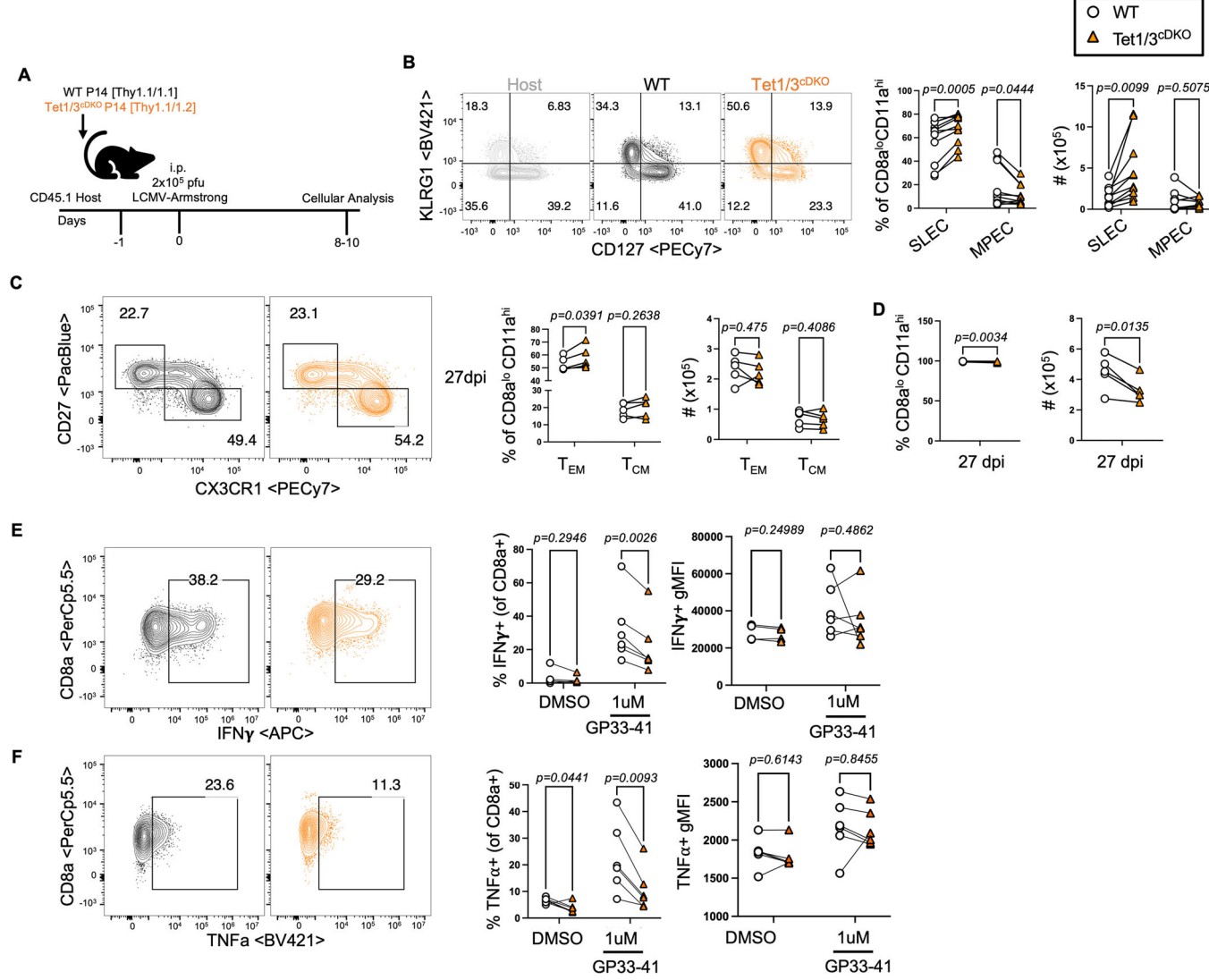

**Figure 2. Tet1/3 restrain the differentiation of SLECs and effector memory T cells in a cell-intrinsic manner.**

In total, 5000 congenically labeled P14 WT (CD90.1/CD90.1) and 5000 P14 Tet1/3cDKO (CD90.1/CD90.2) cells were adoptively co-transferred into naive CD45.1 hosts and infected with $2 \times 10^5$ PFU of LCMV-Armstrong the following day and splenocytes were assessed 8-10dpi or 27dpi (A) Experimental set-up. (B) Representative FACS plots comparing SLEC/MPEC populations from the CD45.1 host, P14 WT, and Tet1/3cDKO and quantification of frequency and number of SLECs/MPECs recovered 8dpi, pregated on LiveTCRb+CD8alo CD11ahi cells. (WT = 8, Tet1/3cDKO = 8 biological replicates). Significance was determined by multiple paired t tests. (C) Representative FACS plots comparing $T_{EM}/T_{CM}$ populations from transferred P14 WT and Tet1/3cDKO and quantification of frequency and number of $T_{EMs}/T_{CMs}$ recovered 27 dpi. Cells were pregated on LiveTCRb+CD8alo CD11ahi cells. (WT = 5, Tet1/3cDKO = 5 biological replicates) Significance was determined by multiple paired t tests. (D) Frequency and number of CD8aloCD11ahi among transferred P14 WT and Tet1/3cDKO 27 dpi. (WT = 5, Tet1/3cDKO = 5 biological replicates) Significance was determined by paired t test. (E) Representative flow plots and frequency of IFNγ+ and IFNγ gMFI among IFNγ + CD8+ T cells following ex vivo gp33-41 peptide restimulation of splenocytes in the presence of Brefeldin A and Monensin for 5 h. Splenocytes were isolated from mice 10 dpi (WT = 6, Tet1/3cDKO = 6). Significance was determined by multiple paired t tests. (F) Representative flow plots and frequency of TNFα+ and TNFα gMFI among TNFα + CD8+ T cells following ex vivo gp33-41 peptide restimulation of splenocytes in the presence of Brefeldin A and Monensin for 5 h. Splenocytes were isolated from mice 10dpi (WT = 6, Tet1/3cDKO = 6 biological replicates). Significance was determined by multiple paired t tests. Data information: P values are indicated in the figures and P < 0.05 was considered significant, paired male or female mice were used and no gender biases associated with genotypes were observed. (B–D) Experiments were replicated at least three times. Source data are available online for this figure.

## Tet1/3 restrict SLEC/$T_{EM}$ CD8 T-cell differentiation by controlling TCR-dependent epigenetic circuits

We previously found that Tet1/3 were required for the differentiation of thymic regulatory T cells by regulating both the generation of Treg cell progenitors and TCR-dependent IL-2 production in CD4+ thymocytes (Teghanemt et al, 2023). To investigate the cell-

intrinsic mechanisms that promote SLEC and $T_{EM}$ cell differentiation in the absence of Tet1/3, we examined whether TCR-dependent activation of CD8 T cells and subsequent responses were modified in cells lacking Tet1/3. Naive CD8 T cells from Tet1/3cDKO mice were FACS-sorted and stimulated in vitro with anti-CD3/CD28 over a time course. Both WT and Tet1/3cDKO CD8 T cells expressed similar levels of CD69 (Fig. 3A), an early gene

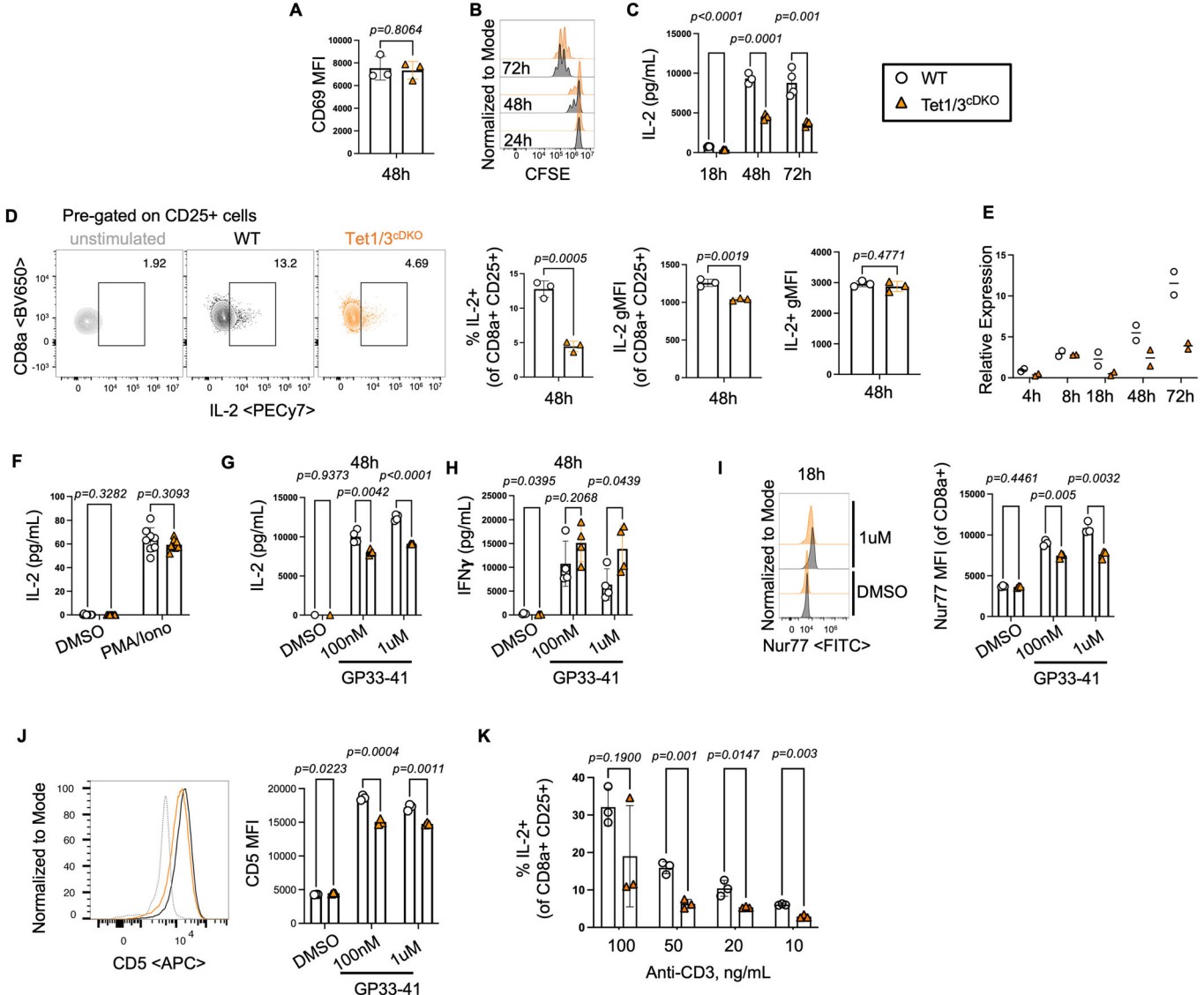

**Figure 3. Tet1/3 restrict SLEC/T_EM CD8 T-cell differentiation by controlling TCR-dependent epigenetic circuits.**

(A–F) FACS-sorted WT and Tet1/3cDKO CD8 T cells were stimulated for indicated time points in vitro using plate-bound anti-CD3 and anti-CD28 (G–J) P14 WT and Tet1/3cDKO splenocytes from naive mice were stimulated in vitro with gp33-41 peptide and anti-CD28 for indicated time points. (A) CD69 MFI on CD8 T cells. (WT = 3, Tet1/3cDKO = 3 biological replicates). Data are mean ± SEM, unpaired t test. (B) Representative histogram showing CD8 T-cell proliferation at indicated time points by CFSE dilution. (C) Quantification of IL-2 levels in the supernatants of activated CD8 T cells by ELISA. (WT = 3-4, Tet1/3cDKO = 4 biological replicates). Data are mean ± SEM, multiple unpaired t tests. (D) Representative FACS plots and quantification of IL-2+ cells, gMFI of IL-2 in bulk CD8 + CD25 + T cells, gMFI of IL-2 among IL-2+ cells, 48-h post activation. Cells were treated with Brefeldin A and Monensin for the last 5 h of stimulation. (WT = 3, Tet1/3cDKO = 3 biological replicates). Data are mean ± SEM, unpaired t test. (E) Relative expression of Il2 normalized to HPRT, by RT-qPCR. (WT = 2, Tet1/3cDKO = 2 biological replicates). (F) Quantification of IL-2 by ELISA in the supernatants of naive CD8 T-cell stimulated with PMA/Ionomycin for 5 h (WT = 8, Tet1/3cDKO = 8 biological replicates). Data are mean ± SEM, multiple unpaired t tests. (G) Quantification of IL-2 by ELISA from supernatants (WT = 4, Tet1/3cDKO = 4 biological replicates). Data are mean ± SEM, multiple unpaired t tests. (H) Quantification of IFNγ by ELISA from supernatants (WT = 4, Tet1/3cDKO = 4 biological replicates). Data are mean ± SEM, multiple unpaired t tests. (I) Representative histogram and quantification of Nur77 MFI among LiveTCRb+CD8a + CD69+ cells. (WT = 3, Tet1/3cDKO = 3 biological replicates). Data are mean ± SEM, multiple unpaired t tests. (J) Representative histogram and quantification of CD5 MFI among LiveTCRb+CD8a + CD69+ cells. (WT = 3, Tet1/3cDKO = 3 biological replicates). Data are mean ± SEM, multiple unpaired t tests. (K) Frequency of IL-2+ cells among CD8 + CD25 + T cells upon 48 h stimulation with varying doses of anti-CD3 and a constant dose of 500 ng/mL CD28. (WT = 3, Tet1/3cDKO = 3 biological replicates). Data are mean ± SEM, multiple unpaired t tests. Data information: P values are indicated in the figures and P < 0.05 was considered significant, paired male or female mice were used, and no gender biases associated with genotypes were observed. (B–D) Experiments were replicated at least two times. Source data are available online for this figure.

upregulated downstream of TCR stimulation and divided comparably over 72 h (Fig. 3B). However, Il-2 production from Tet1/3cDKO cells was profoundly reduced as assessed by both ELISA and correlated with a significantly lower proportion of IL-2 producing cells as assessed by flow cytometry, although the amount of IL-2 secreted in by IL-2+ Tet1/3cDKO cells was comparable to WT cells (Fig. 3C,D). The reduced frequency of IL-2-producing cells correlated with decreased IL-2 mRNA transcripts in bulk CD8

T cells, upon TCR activation (Fig. 3E). However, short-term stimulation of naive CD8 T cells with PMA and ionomycin, which trigger signaling pathways downstream of the TCR but circumvent the TCR itself (Chatila et al, 1989), led to equivalent IL-2 levels (Fig. 3F), suggesting that the defect in IL-2 production by Tet1/3cDKO cells was dependent on TCR-mediated stimulation. In further support of this notion, stimulation of P14 Tet1/3cDKO CD8 T cells with GP33-41 also led to reduced IL-2 production over time compared to controls (Figs. 3G and EV3A), although they expressed similar CD69 expression and divided equally as WT controls (Fig. EV3B,C). Notably, IFN-γ production was not impaired and showed a trending increase in Tet1/3cDKO T cells (Fig. 3H), suggesting that Tet1/3 deficiency in CD8 T cells regulates selective effector responses downstream of TCR activation. Interestingly, Nr4a1 (also known as Nur77), an early response gene that is rapidly upregulated following TCR signaling (Ashouri and Weiss, 2017; Bending et al, 2018; Moran et al, 2011; Zikherman et al, 2012) and used as a proxy for TCR signaling strength, was significantly reduced in peptide-stimulated Tet1/3cDKO P14 T cells (Fig. 3I). This reduction in TCR signaling strength was also captured by examining CD5 expression levels, which were significantly reduced in Tet1/3cDKO P14 cells (Fig. 3J). However, increasing TCR signaling strength by increasing peptide antigen load or increasing Cd3e stimulation did not overcome the reduced ability of Tet1/3cDKO cells to produce IL-2 (Fig. 3G,K). Upon pMHC engagement, the TCR triggers the activation of Erk, which is activated within minutes of antigen encounter and, over these short timescales, appears to function in an all-or-nothing, "digital" manner and exhibit similar signaling levels regardless of pMHC affinity or dose (Altan-Bonnet and Germain, 2005; Gallagher et al, 2021). Importantly, IL-2 expression in T cells is dependent on the binding of the AP-1 family of transcription factors upon TCR activation, which requires ERK pathway activation (Janulis et al, 1999; Samelson, 2002; Walters et al, 2013). Surprisingly, we did not observe impairments in ERK phosphorylation in Tet1/3cDKO T cells upon TCR activation (Fig. EV3D), suggesting that epigenetic circuits that integrate upstream signals are likely impaired in Tet1/3cDKO cells. We previously showed that the ability of CD4 T cells to make IL-2 was selectively dependent on Tet3 (Teghanemt et al, 2023). As expected, we found that IL-2 production in Tet3-deficient CD8 T cells was profoundly impaired following TCR activation. To our surprise, however, Tet3 cKO mice did not show a significant increase in SLEC or T_EM differentiation 8dpi with LCMV, and the proportion of GP33-specific tetramer cells was not reduced, although we did not perform a refined kinetic assessment of SLEC differentiation. As autocrine IL-2 is not required for effector CD8 T-cell differentiation during primary infection (Toumi et al, 2022) and considering the lack of SLEC/T_EM phenotypes in Tet3 cKO mice, we conclude that Tet3 is not sufficient to restrict SLEC/T_EM CD8 T-cell differentiation and that Tet1 or the combined actions of Tet1 and Tet3 are likely required for this process.

## Loss of Tet1/3 impairs chromatin accessibility at regulatory regions in TCR-responsive genes

To interrogate how Tet1/3 regulate selective TCR-dependent transcriptional programs epigenetically, we assessed the chromatin accessibility status of genes by ATAC-Seq (Buenrostro et al, 2015).

As previously reported in human and murine T cells (Bevington et al, 2016; Wang et al, 2018; Yukawa et al, 2020), early TCR stimulation led to a marked change in chromatin remodeling in a large proportion of genes, with 38,130 differentially accessible peaks detected between activated and naive WT T cells (Fold change >1.5, FDR < 0.05, Dataset EV1). However, the number of accessible chromatin regions in activated Tet1/3cDKO cells was profoundly reduced compared to WT controls (Fig. 4A). The majority of these differential chromatin regions were situated in the intragenic and intergenic regions of genes (Fig. 4B) and were associated with genes involved in regulation of TCR-associated pathways, including *Fyn, Itk, Lck, Cd3e*, and *Il-2* (Fig. 4A). In contrast, Tet1/3 deficiency had a very modest impact on the chromatin landscape of genes in naive CD8 T cells (Fig. 4C,D), suggesting that Tet1/3 were mainly required for promoting accessibility in TCR-responsive regulatory regions. In agreement with this, we found that TCR stimulation led to de novo opening of 8980 chromatin regions in WT cells, and 52% of these regions (4669) remained closed in Tet1/3cDKO T cells (Fig. 4E; Dataset EV2). Homer analysis revealed that motifs belonging to the bZIP, ETS, NFAT, and RUNT family of transcription factors (TF) were significantly enriched in differentially accessible regions between Tet1/3cDKO and WT cells (Fig. 4F), with approx. 94% of all differential peaks containing motifs belonging to at least one of these four families. Interestingly, bZIP TF motifs were greatly co-enriched with ETS TF motifs, although a significant fraction of differential peaks only contained bZIP TF motifs (Fig. 4G). ATAC-Seq profiling in P14 T cells isolated from mice 5 dpi with LCMV-Armstrong also revealed significantly reduced chromatin accessibility in Tet1/3cDKO CD8 T cells compared to WT controls. These differences occurred in TCR-associated genes as well as in TF gene loci previously linked to effector and memory T-cell differentiation, such as *Tcf7 and Runx3* (Fig. 4H, Dataset EV3) (Pais Ferreira et al, 2020; Schauder et al, 2021; Wang et al, 2018; Zhou et al, 2010). Mirroring results derived from in vitro-activated T cells, differentially accessible regions in P14 T cells were also highly enriched in motifs belonging predominantly to the bZIP, ETS, NFAT and RUNT family of TFs (Fig. EV4A). The bZIP family of TFs include the AP-1 family and Bach family members which form heterodimeric complexes at palindromic 12-O-Tetradecanoylphorbol-13-acetate response elements (TRE; 5'-TGA(C/G)TCA-3') (Glover and Harrison, 1995; Turner and Tjian, 1989). In contrast to AP-1 TFs which act as transcriptional activators, Bach proteins act as transcriptional repressors and together these TFs play a central role in coordinating TCR-driven effector programs during infection (Oyake et al, 1996; Richer et al, 2016; Roychoudhuri et al, 2016). As JunB is a critical AP-1 member recruited downstream of TCR stimulation and modulates IL-2 production (Katagiri et al, 2019; Yukawa et al, 2020), we assessed whether Tet1/3 deficiency impaired JunB binding. Genome-wide JunB profiling by Cut and Tag (Kaya-Okur et al, 2019) revealed that 27,601 peaks of the 33,941 detected peaks (FDR < 0.05) had an AP-1 motif, underscoring the reliability of the assay (Dataset EV4). However, JunB binding was largely unaffected in Tet1/3cDKO cells (Fig. 4I), including at a putative upstream IL-2 enhancer element and within differentially accessible sites in the *Fyn* locus (Fig. 4J,K). Of note, JunB binding was undetectable within the *Gapdh* and *Cd3e* genes (Figs. 4L and EV4B). Altogether, these findings suggest that Tet1/3-mediated DNA demethylation are required for chromatin accessibility at regulatory regions of key genes downstream of TCR activation independently of JunB binding.

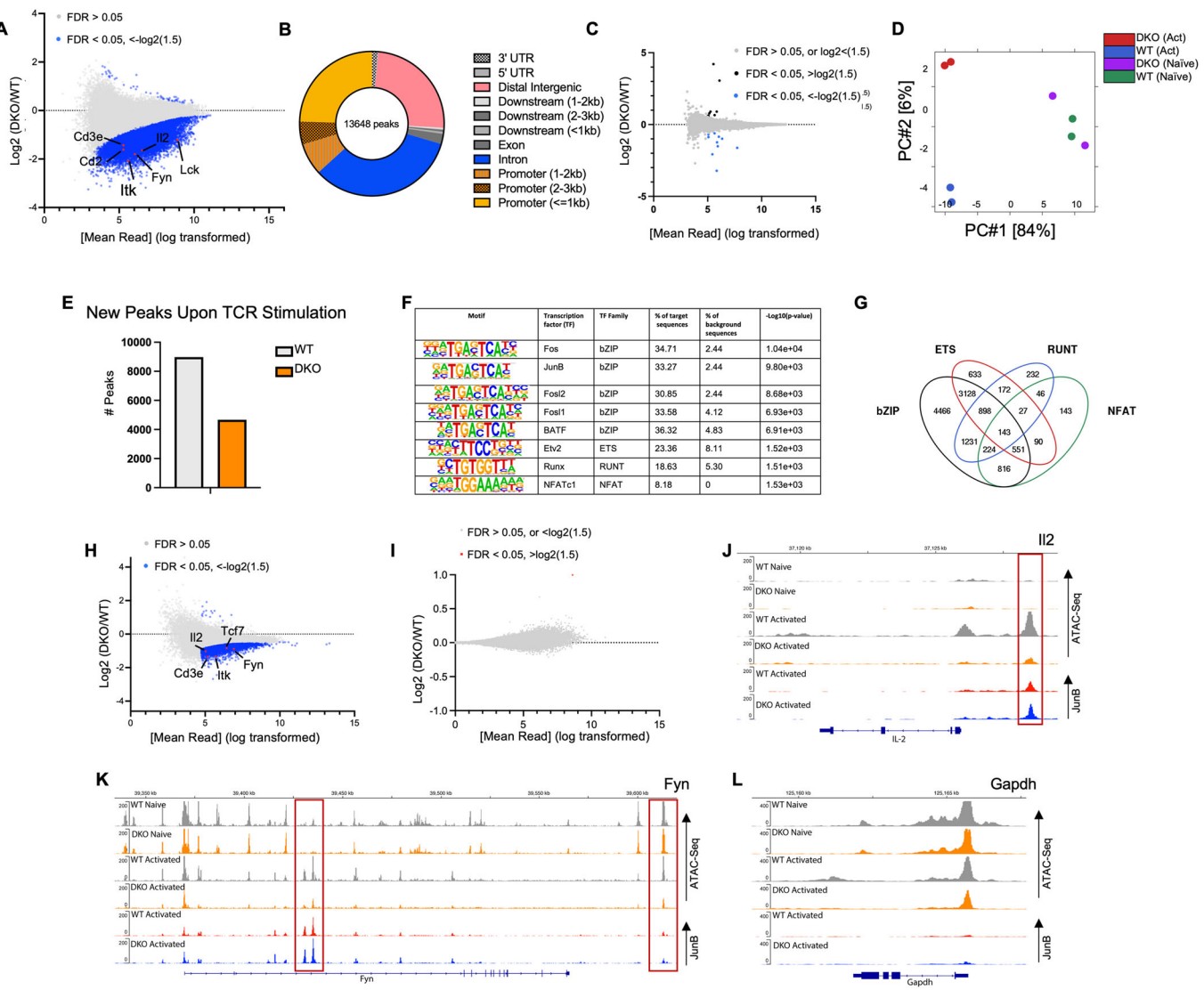

**Figure 4. Loss of Tet1/3 impairs chromatin accessibility at regulatory regions in TCR-response genes.**

(A) Scatter plot depicting regions with differences in chromatin accessibility between activated Tet1/3cDKO and WT CD8 T cells. T cells were activated for 18 h with anti-CD3/CD28, $n = 2$ biological replicates/genotype. (B) Pie Chart depicting the proportions of differentially accessible peaks between activated Tet1/3cDKO and WT CD8 T cells within distinct genomic regions. (C) Scatter plot depicting regions with differences in chromatin accessibility in ex vivo FACs-sorted naive Tet1/3cDKO and WT CD8 T cells, $n = 2$ biological replicates/genotype. (D) PCA plot showing distinct clusters of samples based on genotype and treatment. (E) Bar graph showing the number of de novo accessible peaks that are open following TCR activation in WT T cells and the number of peaks present in Tet1/3cDKO counterparts (FDR < 0.05). (F) Enriched DNA motifs in differentially accessible regions between activated Tet1/3cDKO and WT CD8 T cells. The table shows the percentage of regions with motifs identified by HOMER analysis and selected with an adjusted P value ≤ $10^{-5}$ and target/background >2. (G) Overlap of differentially accessible regions containing bZIP, ETS, Runt or NFAT motifs in in vitro-activated Tet1/3cDKO and WT CD8 T cells. (H) Scatter plot depicting regions with differences in chromatin accessibility between P14 Tet1/3cDKO and WT T cells isolated 5dpi with LCMV-Armstrong, $n = 3$ biological replicates/group. (I) Scatter plot depicting differential JunB peaks identified by Cut and Tag in Tet1/3cDKO and WT T cells 18 h after TCR activation, $n = 3$ biological replicates/group. (J–L) Integrated genome browser view (IGV) shots of the Il-2, Fyn, and Gapdh genes. Tracks show the reference gene, ATAC-Seq peaks (WT in gray, Tet1/3cDKO in orange), and JunB Cut and Tag peaks (WT in red and Tet1/3cDKO in blue) in naive or activated cells.

## CD8 T cells undergo Tet1/3-dependent DNA demethylation during thymic development

Following 4–8 days of LCMV infection, a significant number of genes become hypomethylated in CD8 T cells, particularly in effector gene loci (Scharer et al, 2013; Youngblood et al, 2017). The degree to which this is due to active demethylation rather than passive loss of methylation during T-cell replication and whether early changes in

DNA demethylation dictate cell fate programs remain unclear. We therefore investigated the contribution of Tet1/3-dependent demethylation during early TCR stimulation by examining genome-wide DNA methylation by enzymatic methyl sequencing (EM-Seq) (Vaisvila et al, 2021). Like bisulfite sequencing, EM-Seq does not distinguish between 5mC and 5hmC, reading both as a methylated cytosine residue and are thus referred to as 5mC/5hmC to indicate detection of either. CpG sites with at least 3× coverage were retained in the downstream

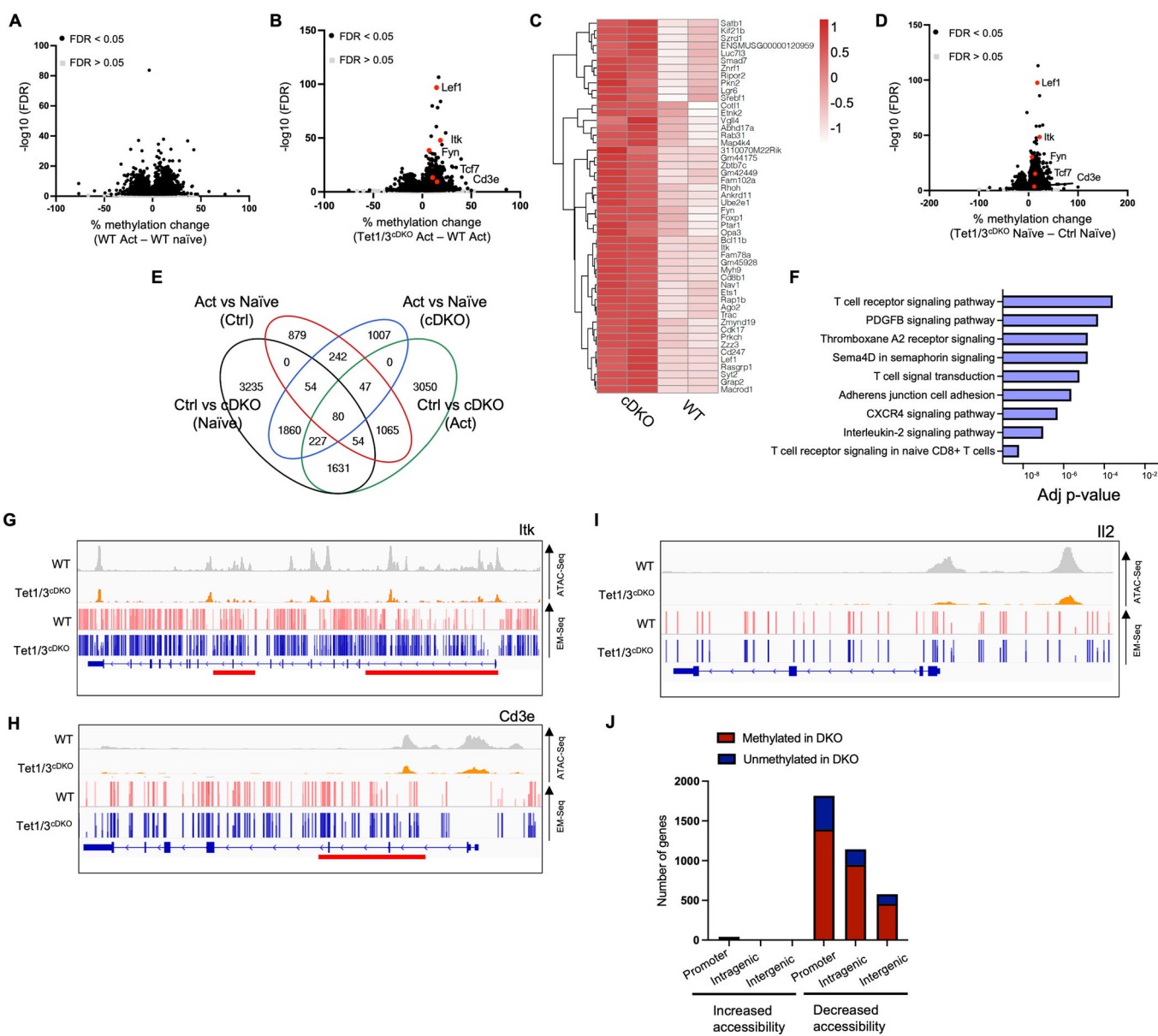

**Figure 5. CD8 T cells undergo Tet1/3-dependent DNA demethylation during thymic development.**

(A) Volcano plot depicting % methylation change in gene loci in WT CD8 T cells upon TCR activation for 18 h, *n* = 2 biological replicates/group. (B) Volcano plot depicting % methylation change in gene loci between activated Tet1/3cDKO and WT CD8 T cells *n* = 2 biological replicates/group. (C) Heatmap representing the top 50 differentially methylated gene loci between activated Tet1/3cDKO and WT CD8 T cells. (D) Volcano plot depicting % methylation change in gene loci between naive Tet1/3cDKO and WT CD8 T cells, *n* = 2 biological replicates/group. (E) Overlap of differentially methylated gene loci in various treatment and genotype comparisons indicated. (F) Gene enrichment analysis of differentially methylated gene loci between naive Tet1/3cDKO and WT CD8 T cells. Adjusted *P* value was calculated using the Benjamini–Hochberg method. and *P* < 0.05 was considered significant. (G–I) Integrated genome browser view (IGV) shots of the Itk, IL-2, and Cd3e genes. Tracks show the reference gene, ATAC-Seq peaks (WT in gray, Tet1/3cDKO in orange), and EM-Seq methylation peaks (WT in red and Tet1/3cDKO in blue) in activated T cells. (J) Bar graph showing the number of gene loci with increased or decreased chromatin accessibility in the promoter, intergenic, or intragenic regions of Tet1/3cDKO cells and having increased or reduced DNA methylation within the gene body and promoter regions (FDR < 0.1). Source data are available online for this figure.

EM-Seq analysis and covered an average of $3.6 \times 10^6$ individual CpGs. 16 h post-anti-CD3/CD28 mediated TCR stimulation, when T cells had not yet undergone cell division (Fig. 3B), we identified 7758 differentially methylated gene loci between WT naive and activated CD8 T cells. Of these, 2421 gene loci showed reduced 5mC/5hmC in activated CD8 T cells compared to naive counterparts (Fig. 5A; Dataset EV5). However, they were not enriched in the regulation of specific

pathways but broadly associated with the regulation of cellular processes such as mitosis (Fig. EV5A). 423 of these loci (12%) also showed reduced 5mC/5hmC when Tet1/3cDKO CD8 T cells were activated, suggesting that these genes lost methylation in a Tet1/3-independent manner following TCR activation. Remarkably, 8484 loci had differential methylation between activated WT and Tet1/3cDKO T cells, of which 6154 loci showed significantly reduced 5mC/5hmC in

WT cells compared to Tet1/3^cDKO (Fig. 5B). The top 50 gene loci included *Itk* and *Fyn* (Fig. 5B,C) which are important mediators of antigen receptor signaling in T cells (Andreotti et al, 2010; Palacios and Weiss, 2004). Other gene loci with significantly reduced methylation in WT cells compared to Tet1/3^cDKO counterparts included *Itk, Cd3e, Tcf7, Lef1,* and *Satb1* (Fig. EV5B). Increased 5mC/5hmC marks in these cells correlated with a significant reduction in mRNA transcripts of these genes in activated Tet1/3^cDKO CD8 T cells compared to WT controls (Fig. EV5C), and we observed lower Cd3e protein expression in Tet1/3^cDKO CD8 T cells (Fig. EV5D). Given that a significantly lower number of genes undergo a loss in methylation upon T-cell activation in WT cells, we tested whether this result was due to pre-existing methylation present in naive Tet1/3^cDKO cells. Indeed, we identified 7141 loci that have higher 5mC/5hmC in Tet1/3^cDKO naive T cells compared to WT naive controls (Fig. 5D,E), suggesting that Tet1/3 are required for shaping the DNA demethylation landscape in mature CD8 T cells. These results are concordant with previous reports that T cells experience a dramatic increase in 5hmC marks in the bodies of genes as they undergo lineage differentiation into CD4 and CD8 T cells from DP precursors in the thymus (Rodriguez et al, 2015; Sellars et al, 2015; Tsagaratou et al, 2014). Strikingly, gene enrichment analysis revealed that genes predominantly enriched in regulatory pathways associated with IL-2 signaling and TCR signaling, such as *Itk, Cd3e, Fyn, Lck* were dependent on the activities of Tet1/3 (Figs. 5F–H and EV5C). Consistent with our previous findings in CD4 T cells (Teghanemt et al, 2023), we did not identify significant differences in methylation proximal or upstream to the IL-2 gene promoter between WT and Tet1/3^cDKO cells (Fig. 5I), indicating that the defect in IL-2 production in Tet1/3 CD8 T cells was likely due to expression defects in modulators of the TCR signaling pathway. We next asked whether there was a correlation between increased 5mC/5hmC marks present *in cis* with changes in chromatin regions in regulatory regions of gene loci in Tet1/3cDKO cells. Indeed, a large number of genes with increased 5mC/5hmC marks present in the gene body+3 kb upstream of the promoter had decreased chromatin accessibility in their promoter, intragenic, and intergenic regions (Fig. 5J), in concordance with a numerous body of literature showing the negative correlation between gene body DNA methylation and chromatin accessibility (Greenberg and Bourc'his, 2019). Taken together, these data support the notion that Tet1/3-mediated demethylation is critical during thymic development to license the chromatin accessibility landscape of genes following TCR activation.

## Tet1/3 are dispensable in peripheral CD8 T cells for effector and memory differentiation

To determine whether the influence of Tet1/3 on CD8 effector fates stems from thymic programming of gene targets regulating effector T-cell differentiation or from their requirement during T-cell activation, we crossed Tet1/3 floxed mice onto an E8iCre-driver, in which Cre activity is observed in peripheral CD8^+ T cells and activated CD8 T cells (Fig. EV6A,B), (Maekawa et al, 2008). We hypothesized that the deletion of Tet1/3 in mature CD8 T cells will not impact effector T-cell fates if thymic programming of the TCR-associated gene landscape is allowed to occur properly during lineage specification of CD8 T cells. We first confirmed genomic excision of Tet1/3 alleles by qPCR and found recombination efficiency of both alleles in sort-purified CD8 T cells from E8iCreTet1/3 floxed mice was as efficient as in sort-purified CD8

T cells from RorcCreTet1/3 floxed mice (Fig. EV6C). As expected, the thymic and peripheral CD4 and CD8 T-cell populations were unchanged in E8iCreTet1/3 floxed mice compared to controls (Fig. EV6D–F). In contrast to observations made in Rorc(t)Cre Tet1/3-deficient CD8 T cells, upon anti-CD3/CD28 mediated TCR stimulation, we observed no difference in IL-2 production in E8iCreTet1/3^fl/fl CD8 T cells compared to controls (Fig. 6A). In all, 8 dpi with LCMV-Armstrong, the proportion and total number of activated CD8 T cells in E8iCreTet1/3 floxed mice were comparable to WT controls (Fig. 6B). Moreover, the proportion of GP33-specific T cells were comparable to WT controls (Fig. EV6G) and no skewed differentiation into SLECs was observed (Fig. 6C) and upon ex vivo stimulation with GP33-41 peptide, E8iCreTet1/3^fl/fl CD8 T cells produced equal amounts of IFNγ as controls (Fig. 6D). Likewise, GP33-specific T_EM cell frequencies and numbers in E8iCreTet1/3 floxed mice was comparable to WT control mice 30 dpi (Figs. 6E,F and EV6H), in sharp contrast to observations made in Rorc(t)Cre Tet1/3 floxed mice (Fig. 1E). We confirmed T_EM/T_CM memory fates via Tbet expression, in addition to surrogate CD27/CX3CR1 markers (Fig. EV6I). Furthermore, the gene expression of genes such as *Fyn, Itk* and *Cd3e* were not impaired in E8iCreTet1/3^fl/fl CD8 T cells (Fig. EV6J), nor was protein expression of TCRβ and Cd3e altered (Fig. EV6K,L) Lastly, to compare thymic programming via 5mC demethylation in developing CD8SP cells, we measured global 5hmC levels in CD8SP thymocytes FACS-purified from RorcCreTet1/3 floxed and E8iCreTet1/3 floxed mice. Consistent with our previous work and work from others (Issuree et al, 2018; Tsagaratou et al, 2014), we found an increase in 5hmC levels in CD8SP cells compared to DP thymocytes from WT mice. However, CD8SP from E8iCreTet1/3 floxed mice had comparable 5hmC to WT cells, while CD8SP cells from Rorc(t)Cre Tet1/3 floxed mice had significantly reduced 5hmC levels (Fig. 6G), in agreement with a critical role of Tet1/3 during lineage differentiation from DP precursors. Taken together, these data suggest that Tet1/3 are dispensable for effector and memory cell fates during peripheral CD8 T-cell priming but are key determinants of these fates via developmental programming.

## Discussion

T cells undergo dynamic changes in DNA methylation in key genes during effector and memory T-cell differentiation following an acute viral infection (Scharer et al, 2013; Youngblood et al, 2017). Understanding the importance of maintaining DNA methylation versus removing DNA methylation in the determination of T-cell lineage formation and function is key to developing context-appropriate therapeutic strategies in the clinic. Currently, our understanding of the role of DNA demethylation on T-cell fate outcomes during an acute infection remains incomplete. The ablation of Tet2 in developing T cells favors MPEC formation and the generation of T_CM cells following acute LCMV infection. Paradoxically however, ablation of Dnmt3a in activated CD8 T cells was also found to enhance the generation of MPECs and T_CM cells during acute LCMV infection (Youngblood et al, 2017). Furthermore, T cells also express Tet1 and Tet3. However, their roles in determining CD8 T-cell fates during acute viral infection remain unknown, adding complexity to our understanding of how the dynamics of methylation and demethylation impact T-cell fates.

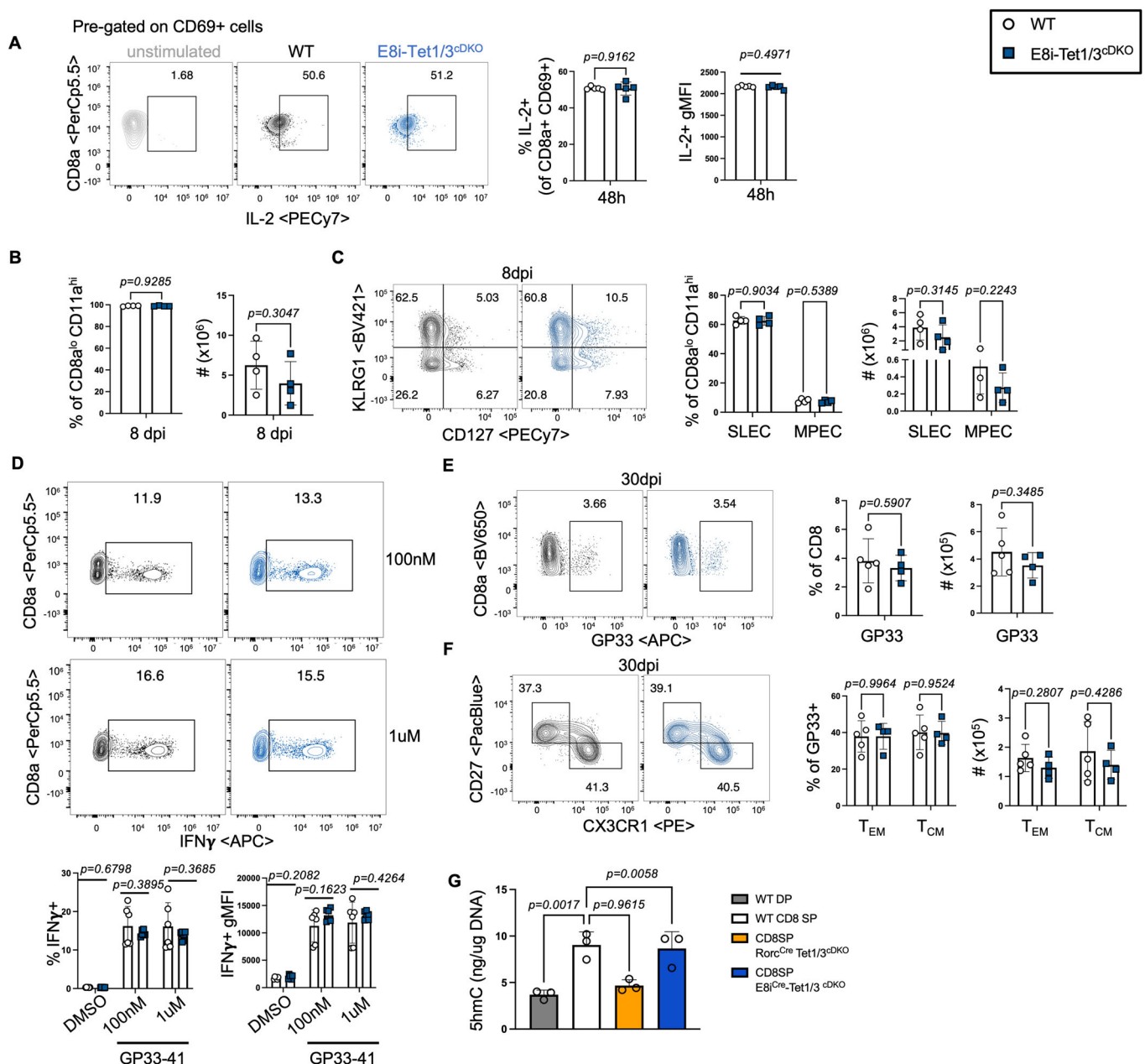

**Figure 6. Tet1/3 are dispensable in peripheral CD8 T cells for effector and memory cell differentiation.**

(A) Representative FACS plots and quantification of frequency of IL-2+ and gMFI of IL-2+ cells, 48 h post activation of FACS-sorted naive T cells with anti-CD3 and anti-CD28. Cells were treated with Brefeldin A and Monensin for the last 5 h of stimulation. (WT = 5, Tet1/3cDKO = 5 biological replicates). Data are mean ± SEM, unpaired *t* test. **(B–F)** WT and E8iCretgTet1/3fl/fl mice were infected with 2 × 10⁵ PFU of LCMV-Armstrong. **(B)** Percentage and number of antigen-experienced CD8 T cells 8dpi. (WT = 4, Tet1/3cDKO = 4 biological replicates) Data are mean ± SEM, unpaired *t* test. **(C)** Representative FACS plots and quantification of frequency and number of SLECs/MPECs from the spleen 8dpi (WT = 4, Tet1/3cDKO = 4 biological replicates) Data are mean ± SEM, multiple unpaired *t* tests. **(D)** Frequency of IFNγ-producing and IFNγ gMFI among IFNγ + CD8 T cells following gp33-41 peptide restimulation of splenocytes, isolated from mice 8dpi. (WT = 6, Tet1/3cDKO = 6 biological replicates). Data are mean ± SEM, multiple unpaired *t* tests. **(E)** Representative FACS plots and quantification of frequency and number of GP33-tetramer+ CD8 T cells in the spleen 30 dpi. (WT = 5, Tet1/3cDKO = 4 biological replicates) Data are mean ± SEM, unpaired *t* test. **(F)** Representative FACS plots and quantification of frequency and number of TEM/TCM, among GP33+ cells. (WT = 5, Tet1/3cDKO = 4 biological replicates) Data are mean ± SEM, multiple unpaired *t* tests. **(G)** ELISA quantification of 5hmC in DNA isolated from the indicated cell types. (WT = 3, Tet1/3cDKO biological replicates/ group). Data are mean ± SEM, One-Way ANOVA test. Data information: *P* values are indicated in the figures and *P* < 0.05 was considered significant; paired male or female mice were used and no gender biases associated with genotypes were observed. **(A–D)** Experiments were replicated at least two times. Source data are available online for this figure.

In this study, we investigated the roles of Tet1/3 on CD8 effector and memory cell fate outcomes during acute LCMV infection. Surprisingly, we found that deletion of Tet1/3 in developing thymocytes resulted in increased generation of SLECs and T$_{EM}$ cells following acute LCMV infection, in contrast to outcomes seen upon the deletion of Tet2 (Carty et al, 2018), underscoring the distinct and non-redundant roles of Tet1/3 in T cells. Remarkably, however, Tet1/3 were dispensable in peripheral CD8 T cells for SLEC/MPEC and T$_{EM}$/T$_{CM}$ cell fates following TCR activation. Instead, Tet1/3 were critically required for the epigenetic programming of a significant number of genes associated with regulation of the TCR-response pathway during thymic development. A failure to undergo DNA demethylation during thymic development did not alter the gross phenotypic characteristics of naive peripheral CD8 T cells. However, TCR stimulation unmasked selective alterations in effector programs in these cells, such as reduced IL-2 production and reduced TCR signaling strength. At a genomic level, T cells deficient in Tet1/3 displayed a significant decrease in chromatin accessibility within genes linked to the regulation of the TCR response as well as memory T-cell differentiation. Crucially, these differentially accessible regions were significantly enriched in motifs belonging to the bZIP TF family. AP-1 factors comprised of cFos and JunB heterodimers have previously been linked to binding almost 70% of early chromatin-accessible regions downstream of TCR signaling in human T cells (Yukawa et al, 2020). However, JunB binding was largely intact in Tet1/3-deficient CD8 T cells following TCR activation, implying that while DNA methylation hinders chromatin accessibility, it does not interfere with JunB binding. These results also suggest that JunB binding is not sufficient for promoting chromatin accessibility downstream of TCR stimulation and that increased binding of Bach factors, which compete for AP-1 binding sites to repress transcription (Jang et al, 2017; Roychoudhuri et al, 2016), are likely not the mechanisms behind reduced chromatin accessibility in activated Tet1/3-deficient T cells. Studies by others delving into how regulatory elements are poised for activation have proposed a model in which AP-1 transcription factors, in collaboration with lineage-specific transcription factors, engage with enhancers initially occluded by nucleosomes. This interaction recruits the SWI/SNF (BAF) remodeling complex, leading to nucleosome remodeling and the establishment of an accessible chromatin state (Vierbuchen et al, 2017; Yukawa et al, 2020). We thus propose a model whereby DNA methylation impedes either the recruitment of the BAF complex by AP-1 factors or interferes with nucleosome remodeling to allow for the activation of TCR-responsive elements that promote gene expression in effector T cells. Further studies are warranted to test this model.

Among key genes that regulate the TCR-response pathway, *Itk*, *Fyn*, and *Cd3e* were critical targets of DNA demethylation during thymic development, showing that the TCR-associated gene landscape is actively regulated by Tet1/3-dependent DNA demethylation. However, a limitation in our present study is that it is unclear whether the propensity of Tet1/3$^{cDKO}$ T cells to differentiate into SLECs and T$_{EM}$ cells is solely a consequence of altered regulation of the TCR-response pathway genes. Strong and sustained IL-2 signals promote effector and repress memory T-cell development (Kalia et al, 2010; Pipkin et al, 2010). However, autocrine IL-2 is dispensable for SLEC and T$_{EM}$ differentiation (Toumi et al, 2022). Furthermore, considering that the co-transfer of WT P14 cells with Tet1/3cDKO cells into WT hosts failed

to "rescue" the skewed SLEC and T$_{EM}$ differentiation and that Tet3 cKO T cells did not show similar skewing despite the defect in IL-2, it is unlikely that autocrine IL-2 is a major driver of these phenotypes. Whether Tet1 plays a role in CD8 T-cell differentiation and the individual epigenetic contributions of Tet1 and Tet3 in this process are important areas of future investigations. Notably, although significant differences in methylation were seen in Tet1/3cDKO T cells, the catalytic-independent functions that these enzymes can have in this process cannot be ruled out (Ketchum et al, 2024; Xue et al, 2016).

Reduced TCR strength in T cells has previously been shown to increase the proportion of Ag-specific CD8 MPECs (Huang et al, 2015; Smith-Garvin et al, 2010). However, in the absence of Tet1/3, it is difficult to predict how net changes in TCR inputs regulated by a combination of various regulators that have both positive and negative feedback functions on TCR signaling affect effector and memory cell outcomes. Furthermore, increased effector memory cell fate outcomes in the absence of Tet1/3 may also be due to differences in transcription factors such as Tcf1 and Runx3 which are known regulators of central memory cell fate (Jeannet et al, 2010; Pais Ferreira et al, 2020; Wang et al, 2018; Zhou et al, 2010). Nevertheless, our findings depict a novel and important epigenetic mechanism by which the TCR-regulatory landscape in peripheral naive T cells is fine-tuned during development and raise exciting questions about the importance of this mechanism in T-cell immunity and tolerance during aging and stress, where dramatic changes in metabolism can lead to epigenetic alterations in developing T cells and have an impact on vaccination outcomes (Moskowitz et al, 2017; Palmer, 2013; Soto-Heredero et al, 2023). Our results using tetramers to examine clonal populations of CD8 T cells during infection hint that Tet1/3 deficiency may alter the TCR repertoire or immunodominance during an acute immune response, although further investigations are necessary. They also suggest that memory T$_{EM}$ and T$_{CM}$ populations generated in the absence of Tet1/3 do not persist for long, as they were significantly reduced 24 dpi. Future studies examining the recall potential of these cells following a secondary immune challenge will shed light on the importance of Tet1/3 in memory T cells.

The combined activities of Tet2 and Tet3 were previously found to be critical for iNKT T-cell differentiation (Tsagaratou et al, 2017), suggesting that Tet2 is required during thymic T-cell development. Consequently, the possibility that Tet2 deletion in developing T cells leads to increased MPECs and T$_{CM}$ cells because of aberrant epigenetic programming in the thymus cannot be discounted. Revisiting the role of Tet2 in peripheral CD8 T-cell differentiation will be important to reconcile the paradoxical observations that loss of either Dnmt3a or Tet2 results in similar cell fate outcomes during acute LCMV infection. Alternatively, it is plausible that Dnmt3a and Tet2 cooperate to regulate distinct transcriptional nodes necessary for MPEC and T$_{CM}$ T-cell differentiation, as a precedent for such cooperative programs exist in other systems (Zhang et al, 2016). Lastly, although we found that Tet1/3 were dispensable for effector and memory cell differentiation, it remains to be seen whether they play a role in regulating recall memory responses. Future investigations examining changes in DNA methylation in memory CD8 T cells lacking Tet1/3 using the E8iCre mouse model system will provide valuable insights into the gene networks influenced by DNA demethylation in this process. These inquiries will be crucial for anticipating the impact of demethylation on shaping the immune gene landscape during vaccination and will open novel avenues that can be harnessed to strategically influence vaccination outcomes.

# Methods

## Reagents and tools table

| Reagent/resource | Reference or source | Identifier or catalog number |
| --- | --- | --- |
| **Experimental models** | | |
| RorcCreTg1Litt/J | Jackson Labs | 22791 |
| Tet1 floxed (C57BL/6J) | Gift from Dr. Aifantis (NYU) | NA |
| Tet3 floxed mice (129 background) | Gift from Dr. Yi Zhang (Harvard) | NA |
| P14Tg (Thy1.2/Thy1.1) | Gift from Dr. Noah Butler (University of Iowa) | NA |
| Rorc(t)Cre Tet1/3 floxed | This paper; Tet1/3 were backcrossed onto C57BL/6 J > 6 generations | NA |
| E8iCreTg | Gift from Dr. Josalyn Cho; available at Jackson Labs | 8766 |
| B6.SJL-CD45.1 | Jackson Labs | 002014 |
| **Antibodies** | | |
| APC-Cy7 anti-mouse CD45.1 (clone A20) | Tonbo | Cat#: 70-0453, RRID:AB_2621497 |
| BV650 anti-mouse CD4 (clone RM4-5) | Biolegend | Cat#: 100555, RRID:AB_2562098 |
| FITC anti-mouse CD8a (clone 53-6.7) | Biolegend | Cat#: 100706, RRID:AB_312745 |
| BV650 anti-mouse CD8a (clone 53.6.7) | Biolegend | Cat#: 100742, RRID:AB_2563056 |
| PerCp5.5 anti-mouse CD8a (clone QA17A07) | Biolegend | Cat#: 155014, RRID:AB_2890703 |
| PECy7 anti-mouse CD8a (clone 53.6.7) | Biolegend | Cat#: 100722, RRID:AB_312760 |
| PerCp5.5 anti-mouse TCRb (clone H57-597) | Tonbo | Cat#: 65-5961, RRID:AB_2621911 |
| vF450 anti-mouse CD24 (clone M1/69) | Tonbo | Cat#: 75-0242 |
| PE anti-mouse CD25 (clone PC61.5) | Biolegend | Cat#: 102007, RRID:AB_312857 |
| PerCp5.5 anti-mouse CD19 (clone 1D.3) | Tonbo | Cat#: 65-0193, RRID:AB_2621887 |
| FITC anti-mouse CD62L (clone MEL-14) | Tonbo | Cat#: 35-0621, RRID:AB_2621697 |
| PE anti-mouse CD62L (clone W18081D) | Biolegend | Cat#: 161204, RRID:AB_2876576 |
| vF500 anti-mouse CD44 (clone IM7) | Tonbo | Cat#: 85-0441 |
| APC anti-mouse CD44 (clone IM7) | Biolegend | Cat#: 103012, RRID:AB_312962 |
| PE anti-mouse CD69 (clone H1.2F3) | Tonbo | Cat#: 50-0691 |
| APC anti-mouse CD11a (clone M17/4) | Biolegend | Cat#: 101120, RRID:AB_2562779 |
| FITC anti-mouse CD11a (clone M17/4) | Biolegend | Cat#: 101106, RRID:AB_312779 |
| PE anti-mouse CD11a (clone M17/4) | Tonbo | Cat#: 50-0111 |
| BV421 anti-mouse KLRG1 (clone 2F1-KLRG1) | Biolegend | Cat#: 75-5893, RRID:AB_2621966 |
| PECy7 anti-mouse CD127 (clone A7R34) | Tonbo | Cat#: 60-1271, RRID:AB_2621859 |
| FITC CD90.1 anti-mouse (clone OX-7) | Biolegend | Cat#: 202503, RRID:AB_314014 |
| PE CD90.2 anti-mouse (clone 30-H12) | Biolegend | Cat#: 105308, RRID:AB_313179 |
| PE/Dazzle 594 anti-mouse CD90.2 (clone 30-H12) | Biolegend | Cat#: 105340, RRID:AB_2632886 |
| eF450 CD90.2 anti-mouse (clone 30-H12) | Invitrogen | Cat#: 48-0902-82, RRID:AB_1272200 |
| PacBlue anti-mouse CD27 (clone LG.3A10) | Biolegend | Cat#: 124218, RRID:AB_2561546 |
| FITC anti-mouse CX3CR1 (clone SA011F11) | Biolegend | Cat#: 149020, RRID:AB_2565703 |
| PE anti-mouse CX3CR1 (clone SA011F11) | Biolegend | Cat#: 153706, RRID:AB_2734221 |
| PECy7 anti-mouse IL-2 (clone JES6-5H3) | Biolegend | Cat#: 503832, RRID:AB_2561750 |
| APC anti-mouse IFN-g (clone XMG1.2) | Tonbo | Cat#: 20-7311, RRID:AB_2621616 |
| BV421 anti-mouse TNFa (cloneMP6-XT22) | Biolegend | Cat#: 506327, RRID:AB_10900823 |
| FITC anti-mouse Nur77 (clone 12.14) | Invitrogen | Cat#: 53-5965-82, RRID:AB_2574429 |
| PE anti-mouse Nur77 (clone 12.14) | Invitrogen | Cat#: 12-5965-82, RRID:AB_1257209 |
| PECy7 anti-mouse Tbet (clone 4B10) | Biolegend | Cat#: 644824, RRID:AB_2561761 |
| PE/Dazzle 594 Ki67 (clone 11F6) | Biolegend | Cat#: 151219, RRID:AB_2910306 |
| AffiniPure Goat Anti-Armenian Hamster IgG | Jackson ImmunoResearch Labs | Cat#: 127-005-099, RRID:AB_2338971 |
| Anti-Mouse CD28 Purified | Tonbo | Cat#: 70-0281-U025 |
| Anti-Mouse CD3 Purified | Tonbo | Cat#: 70-0031-U100 |
| pERK | Cell Signaling | Cat#: 4370 T |
| B-actin | DSHB | Cat#: 224-236-1 |
| JunB | Cell Signaling | Cat#: 3753 T |
| **Oligonucleotides and other sequence-based reagents** | | |
| Hprt Forward: 5′-GTT GGG CTT ACC TCA CTG CT-3′ | This study | |

| Reagent/resource | Reference or source | Identifier or catalog number |
|---|---|---|
| Hprt Reverse: 5'-TCA TCG CTA ATC ACG ACG CT-3' | This study | |
| Cd8a promoter Forward: 5'-TCT GCA AGG GTG CAT TCT CAC TCT-3' | This study | |
| Cd8a promoter Reverse: 5'-AGC TGC AGA CAG AGC TGA TTT CCT-3' | This study | |
| Il2 Forward: 5'-TGA ACT TGG ACC TCT GCG G-3' | This study | |
| Il2 Reverse: 5'-TGT GTT GTC AGA GCC CTT TAG T-3' | This study | |
| Tet1 Forward: 5'-GTC AGG GAG CTC ATG GAG AC-3' | This study | |
| Tet1 Reverse: 5'-CCT GAG AGC TCT TCC CTT CC-3' | This study | |
| Tet2 Forward: 5'-AAC CTG GCT ACT GTC ATT GCT CCA-3' | This study | |
| Tet2 Reverse: 5'-ATC TTC TGC TGG TCT CTG TGG GAA-3' | This study | |
| Tet3 Forward: 5'-AAC GGC TGC AAA TAT GCT CG-3' | This study | |
| Tet3 Reverse: 5'-TCC TCC TCC TTT GGA TTG TCT-3' | This study | |
| Junb Forward: 5'-AGG CAG CTA CTT TTC GGG TC-3' | This study | |
| Junb Reverse: 5'-TTG CTG TTG GGG ACG ATC AA-3' | This study | |
| Itk Forward: 5'-GGT GTC CGA CTT TGG GAT GA-3' | This study | |
| Itk Reverse: 5'-AGG AGA ACA CCT CTG GGG AT-3' | This study | |
| Cd3e Forward: 5'-TGC TAC ACA CCA GCC TCA AAT-3' | This study | |
| Cd3e Reverse: 5'-CAG CAA GCC CAG AGT GAT ACA-3' | This study | |
| Fyn Forward: 5'-AAG CAC GGA CGG AAG ATG AC-3' | This study | |
| Fyn Reverse: 5'-ATG GAG TCA CTG GAG CC AC-3' | This study | |

| Reagent/resource | Reference or source | Identifier or catalog number |
|---|---|---|
| Tcf7 Forward: 5'-CTG TCC CCT TCC TGC GGA T-3' | This study | |
| Tcf7 Reverse: 5'-TGT CCA GGT ACA CCA GAT CCC-3' | This study | |
| Lef1 Forward: 5'-ACT GTC AGG CGA CAC TTC CAT G-3' | This study | |
| Lef1 Reverse: 5'-GTG CTC CTG TTT GAC CTG AGG T-3' | This study | |
| **Chemicals, enzymes, and other reagents** | | |
| SsoAdvanced Universal SYBR Green Supermix | Bio-Rad | Cat#: 1725271 |
| LCMV gp33-41 peptide | GenScript | Cat#: RP20257 |
| CFSE | Invitrogen | Cat#:C34554 |
| Ghost Dye Violet 510 Viability Dye | Tonbo | Cat#: 13-0870-T100 |
| Ghost Dye Red 710 Viability Dye | Tonbo | Cat#: 13-0871-T100 |
| Zombie NIR Dye | Biolegend | Cat#: 423105 |
| BD OptEIA Mouse IFN-g ELISA Set | BD Biosciences | Cat#: 555138 |
| BD OptEIA Mouse IL-2 ELISA Set | BD Biosciences | Cat#: 555148 |
| Dynabeads Untouched CD8 Cells | Invitrogen | Ref#: 11417D |
| Foxp3/Transcription Factor Staining Buffer Kit | Cytek | Ref#: TNB-0607-KIT |
| RNeasy MiniElute Cleanup Kit | Qiagen | Ref#: 74204 |
| iScript cDNA Synthesis Kit | Bio-Rad | Cat#: 1708890 |
| KAPA Express Extract | Roche | Cat#: 50-196-5275 |
| DNeasy Blood & Tissue Kit | Qiagen | Cat#: 69504 |
| NEBNext Enzymatic Methyl-seq Kit | New England Biolabs | Cat#: E7120S |
| ATAC-Seq Kit | Active Motif | Cat#: 53150 |
| CUT&Tag-IT Assay Kit – Cells | Active Motif | Cat#: 53160 |
| **Software** | | |
| Prism8 | Graphpad | NA |
| BioRender | NA | NA |
| Bowtie 2 version 2.4.1 | https://doi.org/10.1038/nmeth.1923 | N/A |
| MACS2 v2.2.7.1 | https://doi.org/10.18129/B9.bioc.MACSr | N/A |
| ChIPseeker v1.28.3 | https://doi.org/10.18129/B9.bioc.ChIPseeker | N/A |

| Reagent/resource | Reference or source | Identifier or catalog number |
|---|---|---|
| ChIPpeakAnno v3.32 | https://doi.org/10.18129/B9.bioc.ChIPpeakAnno | N/A |
| Trimmomatic-v0.35 | https://doi.org/10.1093/bioinformatics/btu170 | N/A |
| HOMER | http://homer.ucsd.edu/homer/motif/ | N/A |
| methylKit version 1.16.1 | https://doi.org/10.18129/B9.bioc.methylKit | N/A |
| DiffBind version 2.10.0 | https://doi.org/10.18129/B9.bioc.DiffBind | N/A |
| Bismark v0.23.1 | https://doi.org/10.1093/bioinformatics/btr167 | N/A |
| SeqMonk v1.48.1 | https://www.bioinformatics.babraham.ac.uk/projects/download.html#seqmonk | N/A |
| Flow jo (v10.8.2) | BeckmanCoulter | N/A |
| Other | | |
| Illumina NovaSeq 6000 | Illumina | |

## Mice

Tet1 floxed (C57BL/6J) and Tet3 floxed mice (129 backcrossed to C57BL/6J) were kindly provided by Dr. Iannis Aifantis (Moran-Crusio et al, 2011) and Dr. Yi Zhang (Shen et al, 2014), respectively. P14Tg (Thy1.2/Thy1.1) transgenic mice on a C57BL/6 background were generously provided by Dr. Noah Butler. These mice were then backcrossed onto RorcCreTg mice (Jax Lab B6.FVB-Tg(RorcCreTg)1Litt/J #022791). Thy1.1/Thy1.1 homozygous P14-RorcCreTg Tet1/3 floxed mice were generated and maintained in-house. E8iCreTg mice (Jax Lab #008766) were kindly provided by Dr. Josalyn Cho to generate E8iCreTgTet1 and Tet3 floxed mice. B6.SJL-CD45.1 mice were purchased from the Jackson Laboratory (#002014). All mice were maintained under specific pathogen-free (SPF) conditions at the barrier animal facility at the University of Iowa Carver College of Medicine on a 12-h light cycle at 30–70% humidity and temperature of 20–26 °C, with access to standard chow and water. Littermate controls (No Cre Tet1/3 floxed), and sex-matched 6-8-week-old mice were used for all experiments unless specified. $CO_2$ exposure was used as the method for euthanasia for all experiments. Permission for all animal experiments performed was granted by the IACUC at the University of Iowa Carver College of Medicine.

## Naive T-cell sorting

Single-cell suspensions were prepared from the spleen and/or lymph nodes after mashing through 70 μm cell strainers. Red blood cells were lysed in ACK lysis Buffer (Gibco), and a Dynabeads Untouched Mouse T-cell kit (Thermo Fisher) was used to enrich for CD4+ and CD8+ T cells. Cells were surface-stained in IMDM containing 2% FBS for 30 min at 4 °C. Cells were washed, and naive CD8 + T cells were then FACS Sorted on a BD FACSAria II or FACSAria Fusion cell sorter by

gating on CD19⁻CD25⁻CD8⁺CD62LʰⁱCD44⁻ T cells. Cell purity was verified and estimated to be >98%.

## Flow cytometry analysis

For immune subset profiling, single-cell suspensions were prepared from the thymus, spleen, peripheral lymph nodes or mesenteric lymph nodes by mashing through 70-μm cell strainers. Red blood cells were lysed in ACK lysis Buffer (Gibco) and counted on a Countess II Automated Cell Counter (Thermo Fisher). Cells were surface-stained in IMDM containing 2% FBS for 30 min at 4 °C or 20 min at RT followed by fixation with a Foxp3/TF Staining Buffer Kit (Tonbo). Cells were then permeabilized and stained for TFs (1 h, RT). For intracellular detection of cytokine production, cells were incubated in the presence of Monensin and Brefeldin A for 5–6 h at 37 °C, followed by staining. Stained cells were analyzed on a Cytoflex flow analyzer or an LSR-II flow analyzer. Flow jo (v10.8.2) was used for all flow cytometry analysis.

## Antibodies

The antibodies used for flow cytometry and cell sorting were purchased from BioLegend, BD Biosciences, Thermo Fisher Scientific, or Tonbo Biosciences and their clone numbers are CD45.1 (A20), CD4 (RM4-5), CD8α (53-6.7), TCRβ (H57-597), CD24 (M1/69), CD25 (PC61.5), CD19 (1D.3) CD62L (MEL-14), CD44 (IM7), CD69 (H1.2F3), CD11a (M17/4), KLRG1 (2F1-KLRG1), CD127 (A7R34), CD90.1 (OX-7), CD90.2 (30-H12), CD27 (LG.3A10), CX3CR1 (SA011F11), IL-2 (JES6-5H3), Nur77 (12.14), IFNγ (XMG1.2), Tbet (4B10), Ki67 (16A8). Ghost dye was used for the exclusion of dead cells. For further details, please see the Reagents and Tools Table.

## Quantitative real-time reverse transcription

DNase I-treated total RNA was prepared from both naive and activated T cells using RNeasy RNA isolation kit (Qiagen) and cDNA was synthesized using an iScript cDNA synthesis kit (Biorad). Tet1 and Tet3 excision efficiency was assessed following lysis using KAPA Express Extract (Roche). Quantitative PCR was performed using SsoAdvanced Universal SYBR Green supermix (Biorad) and a CFX Connect Real-time PCR detection system (Biorad). The following primers were used for qRT-PCR.

## In vitro activation of naive CD8 + T cells (anti-CD3/CD28)

Flat-bottom tissue culture plates were coated with polyclonal goat affinity-purified antibody to hamster IgG (MP Biomedical). FACS-sorted CD19-CD25-CD8+CD62LhiCD44- naive T cells were seeded in T-cell medium [RPMI 1640 (Gibco), 10% heat-inactivated FBS (RγD), 2 mM L-glutamine, 50 μg/ml gentamicin, 1% Penn/Strep, and 50 μM 2-mercaptoethanol (Gibco)] on IgG-bound plates along with 0.1 μg/mL anti-CD3 (Tonbo, clone 17A2) and 0.025 μg/mL anti-CD28 (Tonbo, clone 37.51) antibodies. At indicated timepoints, supernatants were collected for subsequent analysis and cells were mechanically lifted off the plates, allowed to rest for 30 min at 37 °C and surface markers were stained for live assessment by flow cytometry or fixed as described above for

subsequent staining. CFSE labeling for certain experiments was performed prior to seeding on tissue culture plates.

## In vitro activation of P14 T cells

Collagenase D-treated LNs and spleens were mashed through 70-μm cell strainers, red blood cells (RBCs) were lysed in ACK lysis Buffer (Gibco). Antigen presenting cells and P14-RorcCreTg CD8 T cells were seeded in T-cell medium with LCMV gp33-41 peptide (GenScript) and 0.25 μg/mL anti-CD28 (Tonbo, clone 37.51) antibody.

## Adoptive transfer of P14 T cells and LCMV infection

Spleens from P14 mice were mashed through 70 μm cell strainer, RBCs were lysed. An aliquot of cells was surface-stained to characterize the frequency of TCRβ+CD8+ T cells by flow cytometry. Cell numbers were adjusted such that 5000 WT and 5000 Tet1/3$^{cDKO}$ P14 T cells in 200 μL volume were transferred by i.v. injection into CD45.1 recipients. Mice were challenged with a standard dose of LCMV-Armstrong ($2.0 \times 10^5$ PFU) by i.p. injection one day following adoptive T-cell transfer.

## ELISA

IL-2 and IFNγ measurements in the supernatants were performed using a BD OptEIA mouse IL-2 ELISA Set and BD OptEIA mouse IFNγ ELISA Set (BD Biosciences) following the manufacturer's instructions. Supernatants from anti-CD3/anti-CD28 and gp33-41 peptide-stimulated T cells were diluted between 10 and 100-fold in diluent prior to the ELISA, and undiluted supernatants from unstimulated T cells were used as controls. ELISAs were read at OD450 on SpectraMax 384 Plus spectrophotometer. Cytokine levels were calculated through extrapolation from a standard curve using Excel. 5hmC ELISA was performed using the Global 5hmC DNA ELISA kit (Active Motif), and 25 ng of DNA/sample was used. Technical duplicates were run for each sample.

## Whole-genome enzymatic methyl sequencing (EM-Seq) sample processing

Genomic DNA was isolated from naive sorted cells or from sorted CD8 T cells stimulated for 18 h in vitro using a DNA Extraction Kit (Qiagen) and sonicated to generate fragments of ~350 to 400 base pairs using the Bioruptor Pico sonication device (Diagenode). Unmethylated cytosines were converted to uracils, and sequencing libraries were created using the NEBnext Enzymatic Methyl Seq Kit (New England Biolabs) according to the manufacturer's instructions. DNA libraries were sequenced using an Illumina NovaSeq 6000 system following the manufacturer's protocols.

## Genome-wide methylation data analysis

Sequencing data quality was assessed using FastQC v0.11.4. Adapters were trimmed from the sequencing reads using Trimmomatic-v0.39 (Bolger et al, 2014) using options (java -jar trimmomatic-0.39.jar PE -phred33 $FORWARD_READS$REVERSE_READSIILLUMINACLIP:Nextera-PE:2:30:10:2:keepBothReads LEADING:3 TRAILING:3 AVGQUAL:20 MINLEN:35). Alignment to the GRCm39/mm39 reference genome was performed using Bismark v0.23.1 (Krueger and Andrews, 2011) with options (bismark GRCm39 -1 $FORWARD_READS -2

$REVERSE_READS). Deduplication was performed with deduplicate_bismark (deduplicate_bismark $BISMARK_ALIGNED_BAM). Library quality was assessed based on the percentage of reads that aligned to the genome. Library quality was considered sufficient if greater than 50% of reads uniquely aligned to the genome. Enzymatic methyl conversion efficiency was assessed by evaluating the percent of methylation observed in the CHH genome context. Enzymatic methyl conversion was considered sufficient when this value was less than 3%. Genome coverage was assessed using the bedtools genomecov software v2.25.0. Library genome coverage was considered sufficient if 80% of the genome had a depth of at least 10 reads. For each library that met these quality metrics, methylation percentages at individual CpG positions in the reference genome were quantified using the Bismark Methylation Extractor program (Krueger and Andrews, 2011) with options (Bismark-0.23.1/ bismark_methylation_extractor --bedGraph --paired-end $BISMARK_-DEDUPLICATED_BAM). Differentially methylated loci among the datasets were detected using SeqMonk v1.48.1, and each "Probe" was defined to be the gene body +3 kb upstream ("promoter region") (www.bioinformatics.babraham.ac.uk). Differential methylation analysis was then performed via logistic regression using the R package methylKit version 1.16.1 (Akalin et al, 2012). Visualization of CpG positions on colored heatmaps (white-red) reflects the percent methylated from 0 to 100%. Individual genomic loci were displayed using UCSC Genome Browser (52). Enrichment analysis to identify pathways and transcription factors associated with hypermethylated genes was performed using Enrichr (Xie et al, 2021).

## ATAC-Seq sample processing

ATAC-Seq was performed using ~90,000 cells/sample using an ATAC-Seq Kit (Active Motif) following the manufacturer's protocol. Naive CD8 T cells were sorted and stimulated as described above for 18 h. T cells were mechanically lifted and allowed to rest for 20 min prior to processing for ATAC-Seq. P14 CD8 T cells were harvested 5 dpi for ATAC-Seq by sorting on CD45.1-CD8a + CD11a + CD90.1/CD90.2 cells. After purification of tagmented DNA, a quantitative qPCR determined that on average 8 cycles were required for library amplification. The library was purified, and size was selected with Agencourt AMPure beads to remove >2000-bp fragments and excess primers. Samples underwent quality control and quantification on an Agilent BioAnalyzer by the Genomics core at the University of Iowa before pooling and sequencing on an Illumina Novaseq 6000.

## Cut and Tag sample processing

Activated T cells were harvested following 18 h stimulation, counted, and centrifuged for 3 min at 600 × g at room temperature, and 300,000–500,000 cells per condition were used. Cells were processed for Cut and Tag using a Cut and Tag kit (Active Motif) using the manufacturer's protocol. A commercially available Rabbit JunB antibody (Clone C37F9) from Cell Signaling was used. Libraries were purified, and size selection was performed with Agencourt AMPure beads to remove >2000-bp fragments and excess primers.

## ATAC-Seq and Cut and Tag data analysis

Sequencing adapters were removed from paired-end reads using Trimmomatic version 0.39 (Bolger et al, 2014). Trimmed reads

were aligned to mouse reference genome GRCm38/mm10 using Bowtie 2 version 2.4.1 with options --very-sensitive -X 2000. Following genome alignment, Picard was used to mark PCR duplicates, and Samtools version 1.3.1 was used to remove PCR duplicates, discordant pairs, and alignments to the mitochondrial genome. Peak calling was performed using MACS2 version 2.2.7.1, and differential accessibility analyses of MACS2 peaks were performed using the R package DiffBind version 2.10.0. Differentially accessible peaks were annotated to genome regions of interest using the R package ChIPseeker version 1.18.0, and sequence motif enrichment analyses were performed using HOMER version 4.11.1. A cut-off of 200 bp from the peak detected was used for motif enrichment analysis. Gene ontology analysis was performed using Enrichr (Xie et al, 2021).

## Statistical analysis

Graph Pad Prism (v. 9.0 or 10) was used for all statistical analysis. For comparison between two experimental groups, unpaired *t* test or paired Wilcoxon signed-rank test was used. For multiple comparisons, two-way ANOVA Fishers LSD test were used with indicated multiple comparison on the figure to determine the statistical significance between two specific groups. *P* values of ≤0.05 are considered statistically significant and *P* values are indicated on the figures. All experiments were replicated independently two to three times.

# Data availability

ATAC-seq, Cut and Run, and EM-Seq sequencing files from this work have been deposited to the Gene Expression Omnibus database with accession number GSE252583. Files associated with the main figures of the manuscript have been deposited on DataDryad at https://datadryad.org/share/LkVxTE1twDjO50Vab1hhZJsblYgfDW_uPpJmZEeA02I.

The source data of this paper are collected in the following database record: biostudies:S-SCDT-10_1038-S44319-025-00439-z.

# Peer review information

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

## Acknowledgements

The authors would like to thank Dr. John Harty and Dr. Ryan Zander (University of Iowa) for providing valuable discussion and technical advice and sharing reagents. The authors thank members at the Flow Cytometry Facility, which is a Carver College of Medicine/Holden Comprehensive Cancer Center core research facility at the University of Iowa. The facility is funded through user fees and the generous financial support of the Carver College of Medicine, Holden Comprehensive Cancer Center, and Iowa City Veteran's Administration Medical Center. The authors thank Dr. Y Zhang and Dr. I Aifantis for sharing Tet-mutant mice. This work was supported by NIGMS grant 1R35GM154831-01 (to PDI), the Cystic Fibrosis Foundation University of Iowa RDP Bioinformatics Core, NIH grants 1R01HL163024 (to AAP), R01125446, R01127481, R01163058 (to NSB), F31A1164640 (to JTJ), and T32AI007511 (to KMW and JTJ).

## Author contributions

**Kara M Misel-Wuchter**: Data curation; Formal analysis; Validation; Investigation; Visualization; Methodology; Writing—original draft. **Andrew L Thurman**: Software; Formal analysis; Visualization; Methodology; Writing—review and editing. **Jordan T Johnson**: Methodology. **Athmane Teghanemt**: Validation. **Neelam Gautam**: Validation; Investigation. **Alejandro A Pezzulo**: Supervision; Writing—review and editing. **Jennifer R Bermick**: Formal analysis; Methodology; Writing—review and editing. **Noah S Butler**: Supervision; Writing—review and editing. **Priya D Issuree**: Conceptualization; Data curation; Formal analysis; Supervision; Funding acquisition; Methodology; Writing—original draft; Writing—review and editing.

Source data underlying figure panels in this paper may have individual authorship assigned. Where available, figure panel/source data authorship is listed in the following database record: biostudies:S-SCDT-10_1038-S44319-025-00439-z.

## Disclosure and competing interests statement

The authors declare no competing interests.

# Expanded View Figures

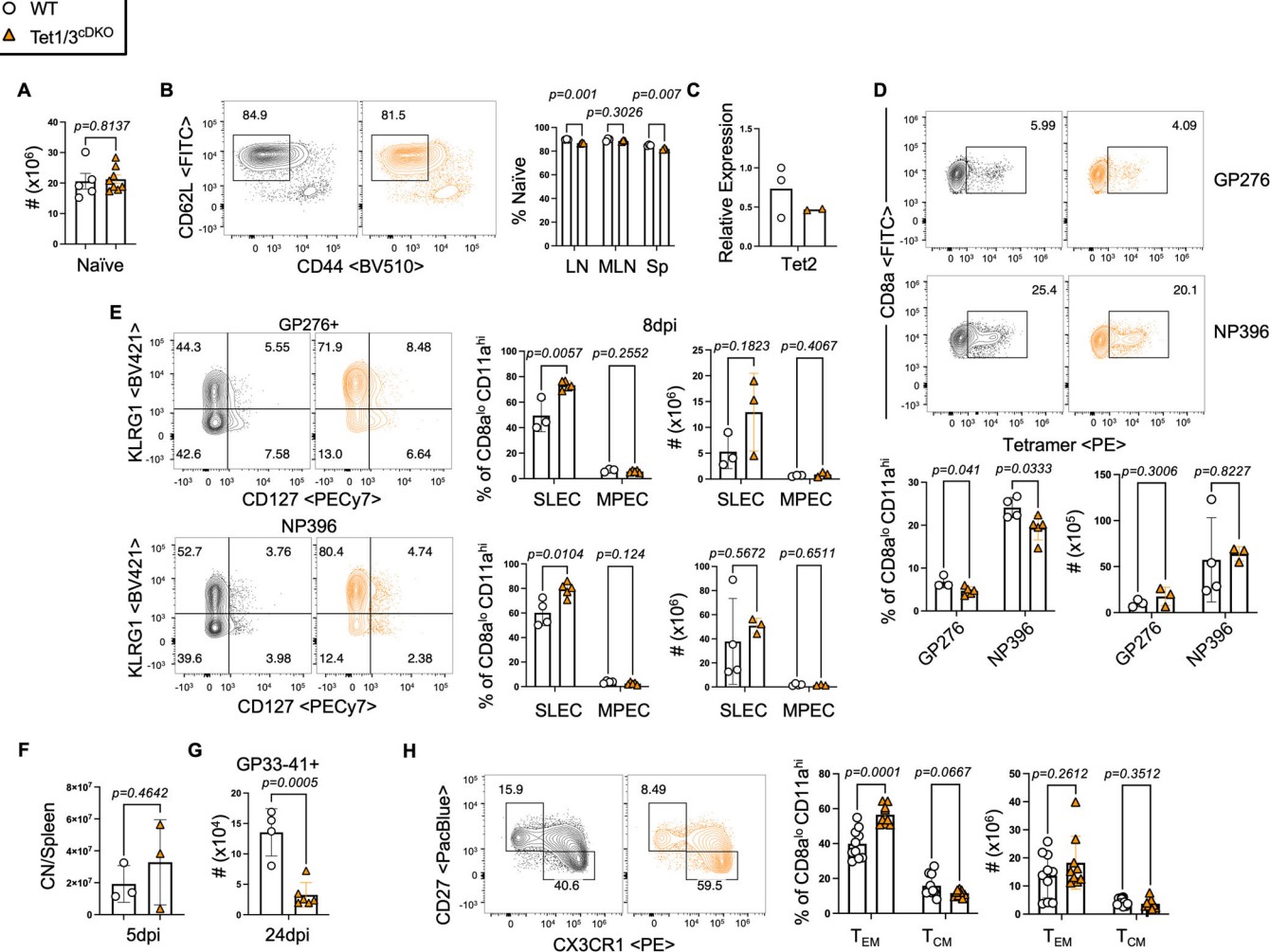

**Figure EV1. Tet1 and Tet3 deficiency results in increased SLECs and effector memory T-cell differentiation during acute LCMV infection.**

(A) Absolute numbers of mature CD8 + T cells from control Tet1/3fl/fl and Rorc(t)CreTgTet1/3fl/fl mice. (WT = 5, Tet1/3^cDKO = 8 biological replicates). Data are mean ± SEM, unpaired t test. (B) Representative FACS plots of CD62L and CD44 expression on CD8 T cells and frequency of naive CD8 T cells from indicated secondary lymphoid organs of control P14 Tet1/3fl/fl and P14-Rorc(t)CreTgTet1/3fl/fl mice, pregated on LiveTCRb+CD8a+ cells. (WT = 3, Tet1/3^cDKO = 3 biological replicates). Data are mean ± SEM, multiple unpaired t tests. (C) Relative expression of Tet2 normalized to HPRT in naive Tet1/3fl/fl and Rorc(t)CreTgTet1/3fl/fl CD8 T cells. (WT = 3, Tet1/3^cDKO = 2 biological replicates). (D) Representative FACS plots and quantification of frequency and number of GP276- and NP396-tetramer+ CD8 T cells. (WT = 3-4, Tet1/3^cDKO = 5 biological replicates) Data are mean ± SEM, multiple unpaired t tests. (E) Representative FACS plots and quantification of frequency and number of SLEC and MPECs among GP276-tetramer+ and NP396-tetramer+ CD8 cells. (WT = 3-4, Tet1/3^cDKO = 5 biological replicates) Data are mean ± SEM, multiple unpaired t tests. (F) RT-qPCR assessment of LCMV copy number (CN) per spleen 5dpi. (WT = 3, Tet1/3^cDKO = 3 biological replicates) Data are mean ± SEM, unpaired t test. (G) Absolute number of GP33-tetramer+ CD8 T cells 24dpi. (WT = 4, Tet1/3^cDKO = 6 biological replicates). Data are mean ± SEM, unpaired t test. (H) Representative FACS and quantification of frequency and number of TEM/TCM 8dpi. (WT = 10, Tet1/3^cDKO = 9 biological replicates). Data are mean ± SEM, multiple unpaired t tests. Data information: P values are indicated in the figures and P < 0.05 was considered significant; paired male or female mice were used and no gender biases associated with genotypes were observed. (A, B, H) Experiments were replicated at least three times. Source data are available online for this figure.

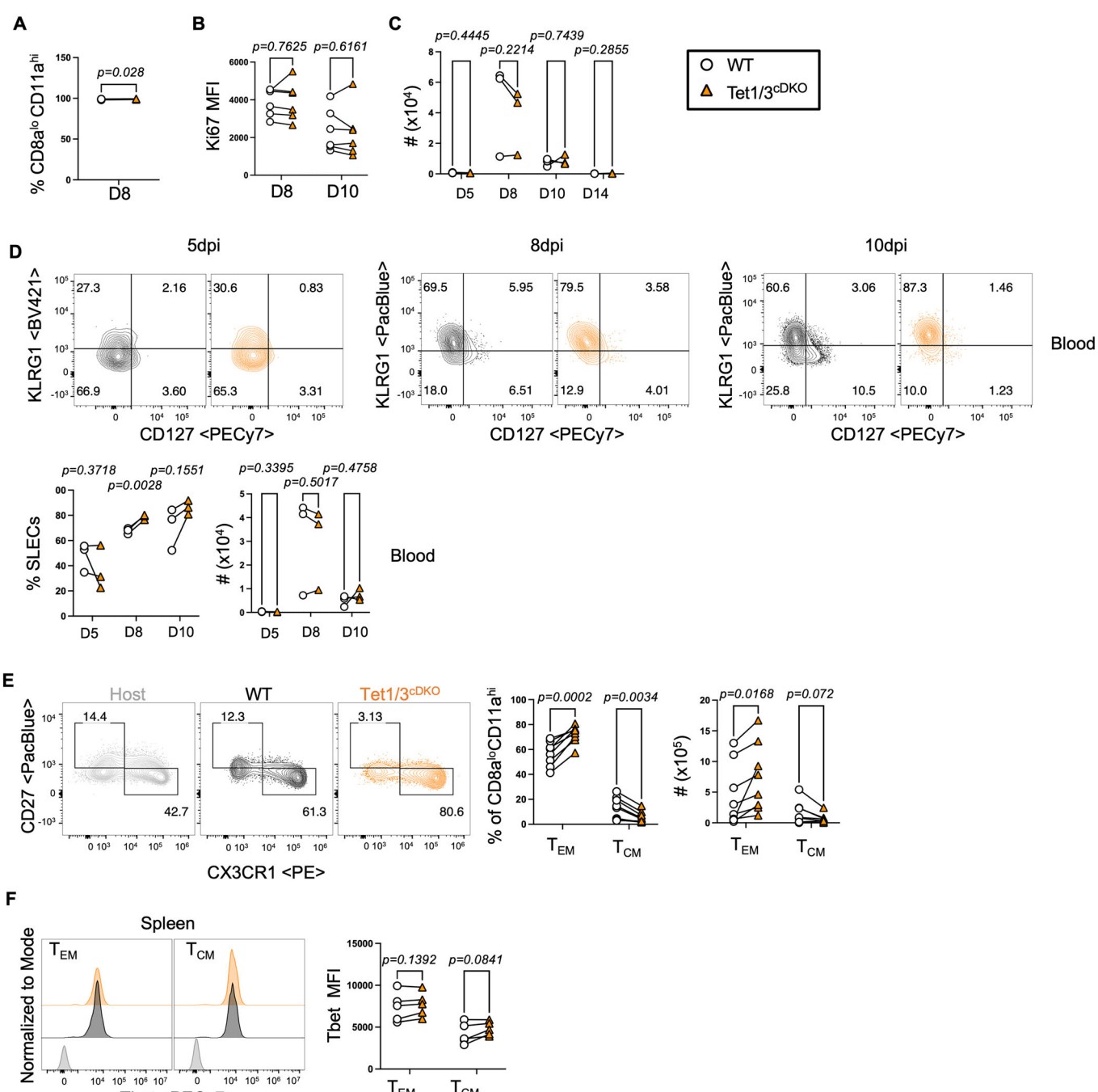

**Figure EV2. Tet1/3 restrain the differentiation of SLECs and effector memory T cells in a cell-intrinsic manner.**

(A) Percent of CD8a^lo^CD11a^hi^ cells among transferred P14 CD8 T cells 8 dpi. (WT = 5, Tet1/3^cDKO^ = 5 biological replicates) Significance was determined by paired *t* test.
(B) Ki67 MFI among transferred P14 CD8 T cells in the blood at 8 and 10 dpi. (WT = 5, Tet1/3^cDKO^ = 5 biological replicates) Significance was determined by multiple paired *t* tests. (C) Absolute numbers of transferred P14 CD8 T cells in the blood at indicated time points. (WT = 3, Tet1/3^cDKO^ = 3 biological replicates) Significance was determined by multiple paired *t* tests. (D) Representative FACS plots and quantification of frequency and number of SLECs recovered at indicated time points in the blood. (WT = 3, Tet1/3^cDKO^ = 3 biological replicates) Significance was determined by multiple paired *t* tests. (E) Representative FACS plots and quantification of frequency and number of TEM/TCM cells recovered 8 dpi. (WT = 8, Tet1/3^cDKO^ = 8 biological replicates) Significance was determined by multiple paired *t* tests. (F) Representative histograms and quantification of Tbet expression in TEM/TCM populations. (WT = 5, Tet1/3^cDKO^ = 5 biological replicates) Significance was determined by multiple paired *t* tests. Data information: P values are indicated in the figures and P < 0.05 was considered significant; paired male or female mice were used and no gender biases associated with genotypes were observed. (A–D) Experiments were replicated at least three times. Source data are available online for this figure.

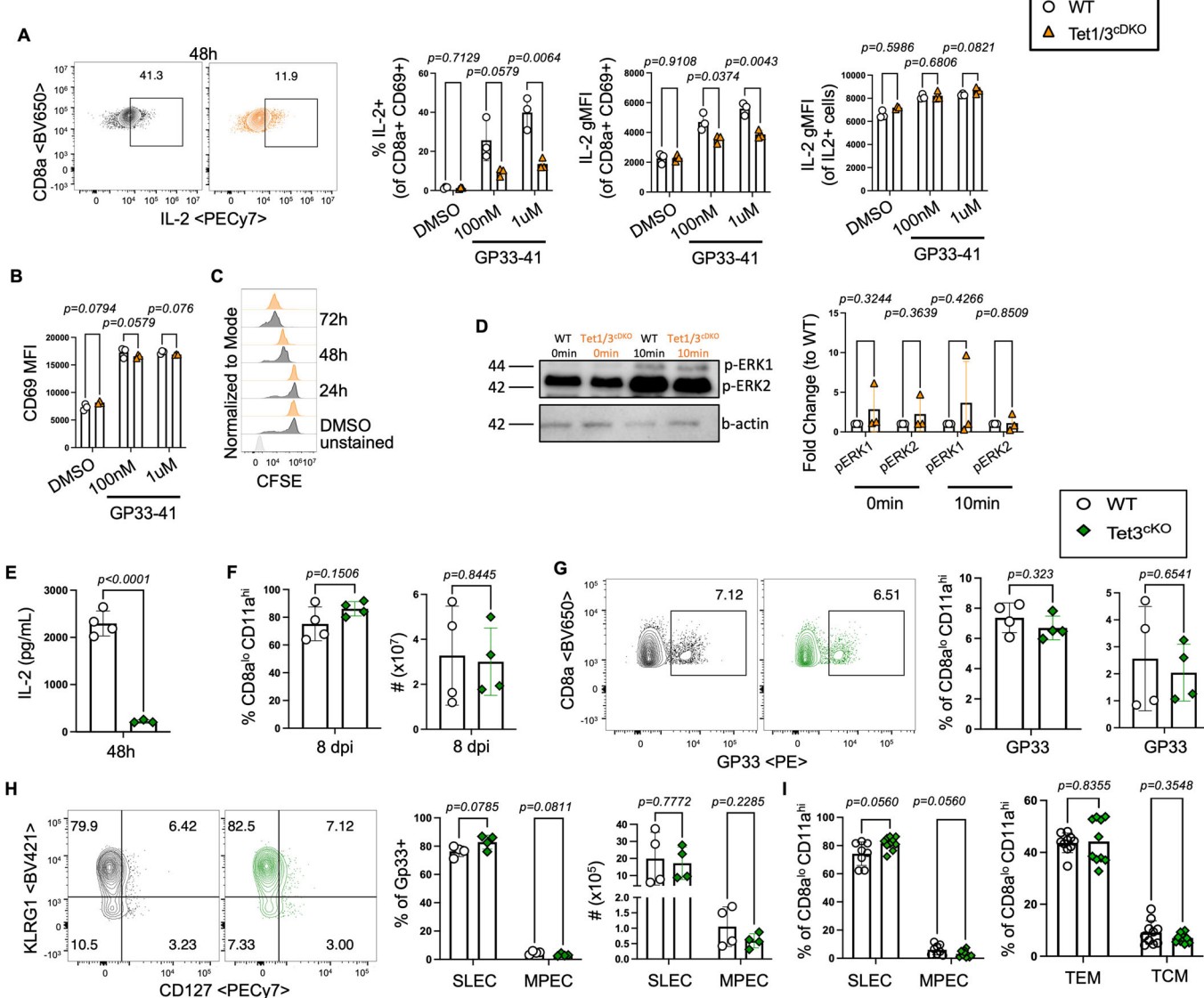

**Figure EV3. Tet1/3 restrict SLEC/T_EM CD8 T-cell differentiation by controlling TCR-dependent epigenetic circuits.**

(A–C) Naive P14 WT and Tet1/3^cDKO splenocytes were stimulated in vitro with gp33-41 peptide and anti-CD28 for indicated time points. (D) FACS-Sorted naive WT and Tet1/3^cDKO T cells were stimulated in vitro via anti-CD3 and anti-CD28 for indicated time points. (A) Representative FACS plots and quantification of frequency of IL-2+ cells among CD69+ cells, gMFI of IL-2 among CD69+ cells, gMFI of IL-2 among IL-2+ cells, 48-h post activation. Cells were treated with Brefeldin A and Monensin for the last 5 h of stimulation (WT = 3, Tet1/3^cDKO = 3 biological replicates). Data are mean ± SEM, multiple unpaired t tests. (B) CD69 MFI on CD8 T cells 48-h post activation (WT = 3, Tet1/3^cDKO = 3 biological replicates). Data are mean ± SEM, multiple unpaired t test. (C) Representative histogram showing CD8 T-cell proliferation by CFSE dilution. (D) Representative immunoblot and quantification of phospho-ERK1/2. Blots were quantified using ImageJ and normalized to show quantification of three independent experiments. Data are mean ± SEM, multiple unpaired t tests. (E) Quantification of IL-2 levels in the supernatants of activated CD8 T cells by ELISA. (WT = 4, Tet3^cKO = 4 biological replicates). Data are mean ± SEM, multiple unpaired t tests. (F) Percentage and number of antigen-experienced CD8 T cells 8dpi. (WT = 4, Tet3^cKO = 4 biological replicates) Data are mean ± SEM, unpaired t test. (G) Representative FACS plots and quantification of frequency and number of GP33-tetramer+ CD8 T cells. Cells were pregated on LiveTCRβ+CD8a^lo CD11a^hi cells. (WT = 4, Tet3^cKO = 4 biological replicates) Data are mean ± SEM, unpaired t test. (H) Representative FACS plots and quantification of frequency and number of SLECs/MPECs among GP33-tetramer+ cells 8 dpi. (WT = 4, Tet3^cKO = 4 biological replicates) Data are mean ± SEM, multiple unpaired t tests. (I) Quantification of frequency of SLECs/MPECs among activated CD8 T cells 8 dpi. Data is a summary of 2 independent experiments (WT = 8, Tet3^cKO = 10 biological replicates) Data are mean ± SEM, multiple unpaired t tests. Data information: P values are indicated in the figures and P < 0.05 was considered significant; paired male or female mice were used and no gender biases associated with genotypes were observed. (A) Experiments were replicated three times. (B–D) Experiments were replicated twice. Source data are available online for this figure.

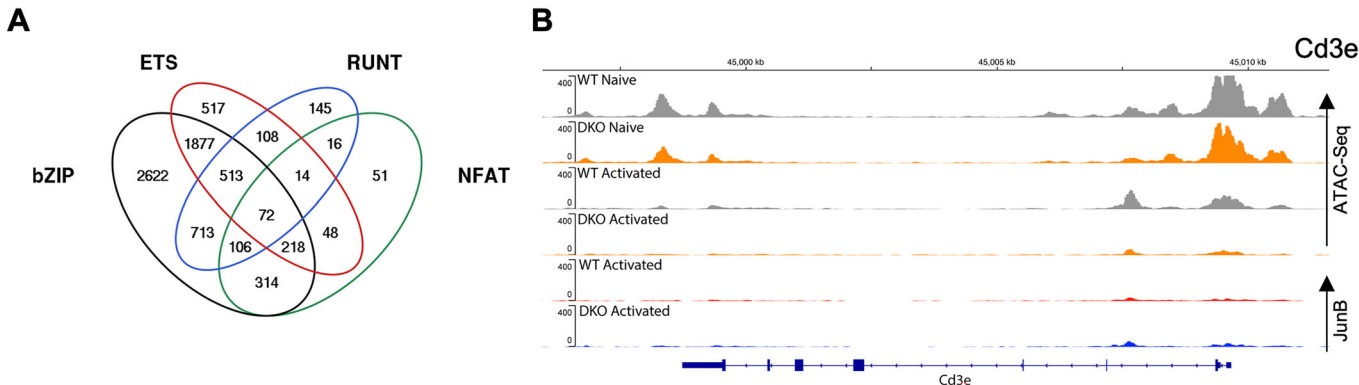

**Figure EV4. Loss of Tet1 and Tet3 impairs chromatin accessibility at regulatory regions in TCR-response genes.**

(A) Overlap of differentially accessible peaks containing bZIP, ETS, Runt or NFAT in P14 Tet1/3cDKO and WT CD8 T cells isolated 5dpi with LCMV-Armstrong (B) Integrated genome browser view (IGV) shots of the Cd3e gene. Tracks show the reference gene, ATAC-Seq peaks (WT in gray, Tet1/3cDKO in orange) and JunB Cut and Run peaks (WT in red and Tet1/3cDKO in blue) in naive or activated cells.

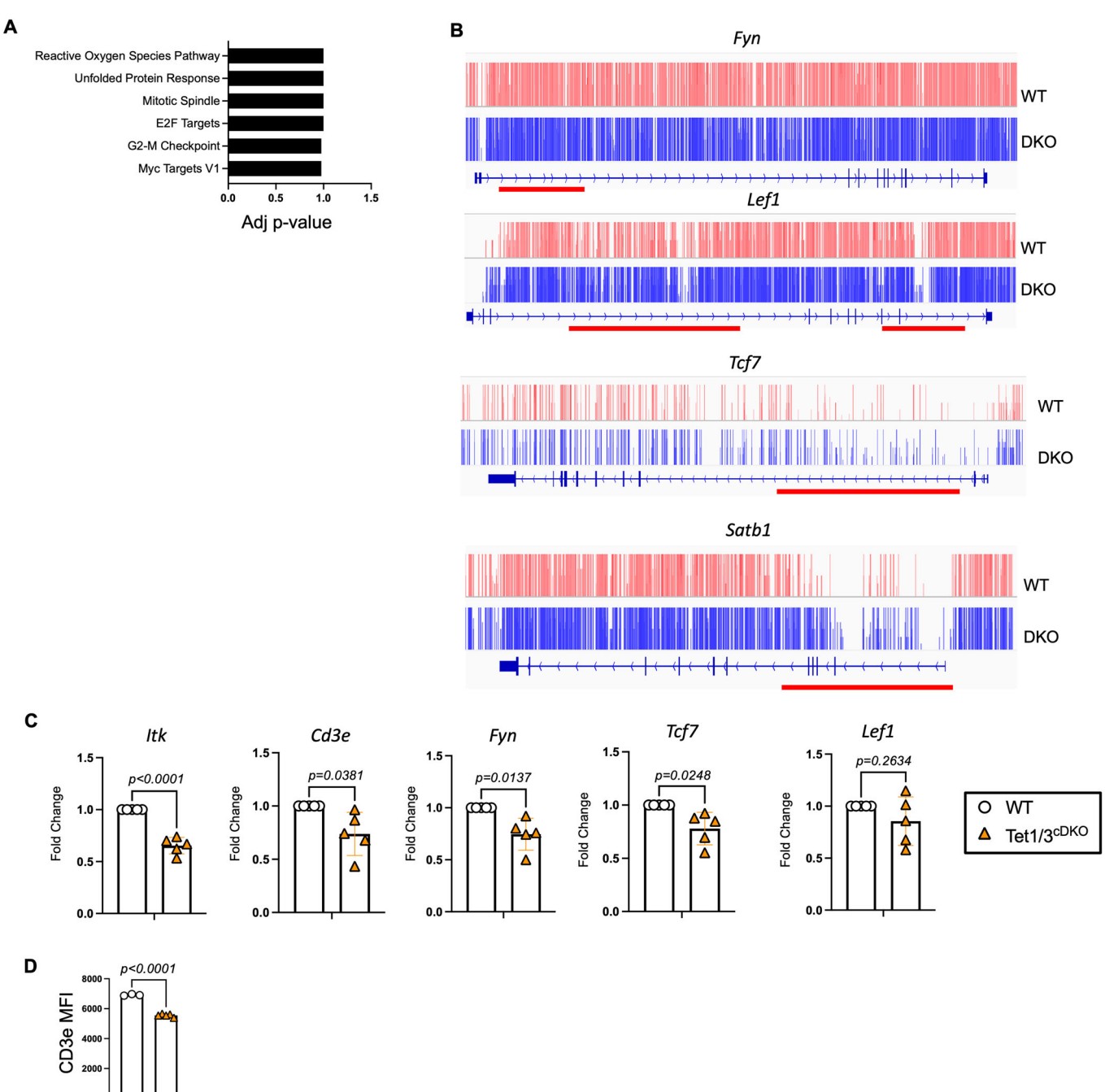

**Figure EV5. CD8 T cells undergo Tet/3-dependent DNA demethylation during thymic development.**

(A) Gene enrichment analysis of differentially methylated gene loci between WT activated and naive CD8 T cells. Adjusted *P* value was calculated using the Benjamini–Hochberg method. (B) Integrated genome browser view (IGV) shots of the Fyn, Lef1, Tcf7 and Satb1 gene loci. Tracks show the reference gene, and EM-Seq methylation peaks (WT in red and Tet1/3cDKO in blue) in activated CD8 T cells. Red bar indicates regions with differentially methylated CpGs. (C) Fold change of Itk, Cd3e, Fyn, Tcf7 and Lef1 expression in activated CD8 T cells by RT-qPCR. Data is normalized to HPRT. (WT = 5, Tet1/3cDKO = 5 biological replicates). Cells were activated in vitro for 48 h with anti-CD3/CD28. Data are mean ± SEM, unpaired *t* test. (D) CD3e MFI 48 h post activation with anti-CD3/CD28. (WT = 3, Tet1/3cDKO = 5 biological replicates). Data are mean ± SEM, unpaired *t* test. Data information: *P* values are indicated in the figures and *P* < 0.05 was considered significant; paired male or female mice were used and no gender biases associated with genotypes were observed. (A) Experiments were replicated three times. (C, D) Experiments were replicated at least three times. Source data are available online for this figure.

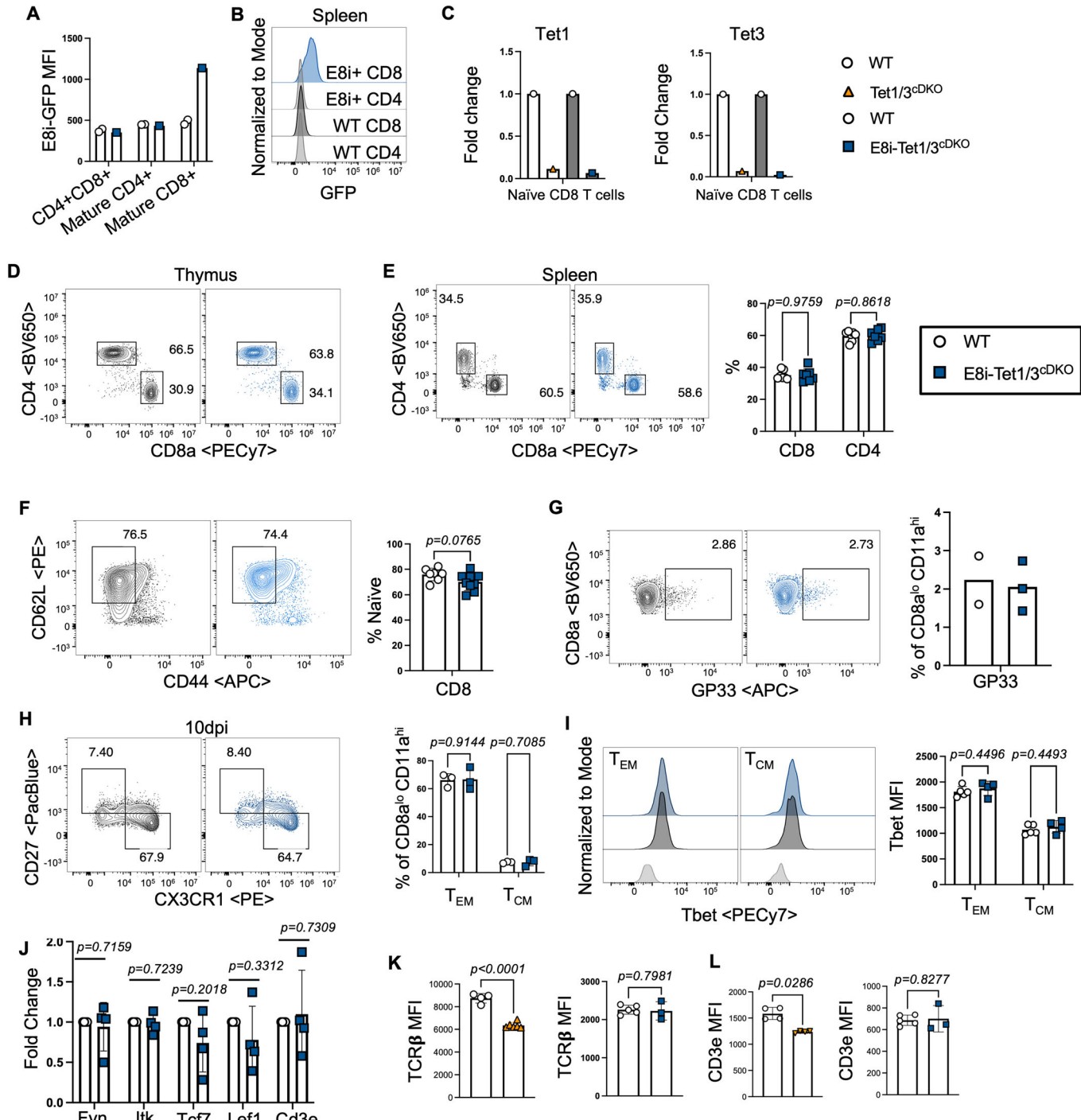

◀ **Figure EV6.  Tet1/3 are dispensable in peripheral CD8 T cells for effector and memory differentiation.**

(A) GFP MFI expressed in control Tet1/3fl/fl and E8iCreTgTet1/3fl/fl thymic populations. (WT = 2, Tet1/3$^{cDKO}$ = 2 biological replicates). (B) Representative histograms showing GFP expression in control Tet1/3fl/fl- and E8iCreTgTet1/3fl/fl lymphocytes from the spleen (C) Fold change of Tet1 and Tet3 genomic copies in naive CD8 T cells from control Tet1/3fl/fl, Rorc(t)CreTgTet1/3fl/fl and E8iCreTgTet1/3fl/fl mice. (D) Representative FACS plots of CD4 and CD8 thymic T-cell populations from control Tet1/3fl/fl and E8iCreTgTet1/3fl/fl mice. Cells were pregated on LiveCD24-TCRb+CD69- cells. (E) Representative FACS plots and frequency of peripheral CD4 and CD8 splenic T-cell populations from control Tet1/3fl/fl and E8iCreTgTet1/3fl/fl mice. Cells were pregated on LiveTCRb+ cells. (WT = 7, Tet1/3$^{cDKO}$ = 7 biological replicates). Data are mean ± SEM, multiple unpaired t tests. (F) Representative FACS plots and frequency of naive CD8 T cells from the lymph nodes and spleens of control Tet1/3fl/fl and E8iCreTgTet1/3fl/fl mice. Cells were pregated on LiveTCRb+CD8a+ cells. (WT = 7, Tet1/3$^{cDKO}$ = 9 biological replicates). Data are mean ± SEM, unpaired t test. (G) Representative histograms and quantification of GP33-tetramer+ CD8 T cells 8 dpi. (WT = 2, Tet1/3$^{cDKO}$ = 3 biological replicates). (H) Representation flow plots and quantification of TEM and TCM populations in the spleen 10 dpi. (WT = 3, Tet1/3$^{cDKO}$ = 3 biological replicates) Data are mean ± SEM, multiple unpaired t tests. (I) Representative histograms and quantification of Tbet expression in TEM/TCM populations 30 dpi. (WT = 5, Tet1/3$^{cDKO}$ = 4 biological replicates) Data are mean ± SEM, multiple unpaired tests. (J) Fold change of Fyn, Itk, Tcf, Lef1, Cd3e mRNA expression in activated CD8 T cells from control and E8iCreTgTet1/3fl/fl mice. Data is a summary of 2 independent experiments (WT = 4, Tet1/3cDKO = 4 biological replicates) Data are mean ± SEM, multiple unpaired t tests. (K) TCRb MFI on bulk CD8 + T cells from the spleen 8 dpi with LCMV-Armstrong. Data are mean ± SEM, unpaired t test. (WT = 4, RorcCreTet1/3$^{cDKO}$ = 6 biological replicates; WT = 5, E8iCre-Tet1/3 $^{cDKO}$ = 3 biological replicates) Data are mean ± SEM, unpaired t test. (L) Cd3e MFI on bulk CD8 + T cells from the spleen 10 dpi with LCMV-Armstrong. Data are mean ± SEM, unpaired t test. (WT = 4, RorcCreTet1/3 $^{cDKO}$ = 4 biological replicates; WT = 5, E8iCre-Tet1/3 $^{cDKO}$ = 3 biological replicates) Data are mean ± SEM, unpaired t test. Data information: P values are indicated in the figures and P < 0.05 was considered significant; paired male or female mice were used and no gender biases associated with genotypes were observed. (A, B, D–L) Experiments were replicated at least two times. Source data are available online for this figure.

                                                                                                                                                                                                                                                                   