## [Peer Review File · EMBO Reports]

Developmental Epigenetic Programming by Tet1/3 is a Critical Determinant of Peripheral CD8 T Cell Fate

Priya Issuree, Kara Misel-Wuchter, Andrew Thurman, Jordan Johnson, Athmane Teghanemt, Neelam Gautam, Alejandro Pezzulo, Jennifer Bermick, and Noah Butler

Corresponding author(s): Priya Issuree (priya-issuree@uiowa.edu)

Review Timeline:

Submission Date:	21st May 24
Editorial Decision:	3rd Jul 24
Revision Received:	30th Jan 25
Editorial Decision:	27th Feb 25
Revision Received:	18th Mar 25
Accepted:	20th Mar 25

Editor: Achim Breiling

Transaction Report:

Dear Dr. Issuree,

Thank you for the submission of your manuscript to EMBO reports. I have now received the reports from the three referees that were asked to evaluate your study, which can be found at the end of this email. As you will see, the referees have several comments, concerns, and suggestions, indicating that a major revision of the manuscript is necessary to allow publication of the study in EMBO reports. As the reports are below, and all the concerns need to be addressed, I will not detail them further here.

Given the constructive referee comments, I would like to invite you to revise your manuscript with the understanding that the concerns of the referees must be addressed in the revised manuscript or in a detailed point-by-point response. Acceptance of your manuscript will depend on a positive outcome of a second round of review. It is EMBO reports policy to allow a single round of revision only and acceptance of the manuscript will therefore depend on the completeness of your responses included in the next, final version of the manuscript.

1) a .docx formatted version of the final manuscript text (including legends for main figures, EV figures and tables), but without the figures included. Figure legends should be compiled at the end of the manuscript text.

2) individual production quality figure files as .eps, .tif, .jpg (one file per figure), of main figures and EV figures. Please upload these as separate, individual files upon re-submission.

4) a complete author checklist, which you can download from our author guidelines

(<https://www.embopress.org/page/journal/14693178/authorguide>). Please insert page numbers in the checklist to indicate where the requested information can be found in the manuscript. The completed author checklist will also be part of the RPF.

5) that primary datasets produced in this study (e.g. RNA-seq, ChIP-seq, structural and array data) are deposited in an appropriate public database. If no primary datasets have been deposited, please also state this in a dedicated section (e.g. 'No primary datasets have been generated and deposited'), see below.

The accession numbers and database should be listed in a formal "Data Availability" section (placed after Materials & Methods) that follows the model below. This is now mandatory (like the COI statement). Please note that the Data Availability Section is restricted to new primary data that are part of this study. This section is mandatory. As indicated above, if no primary datasets have been deposited, please state this in this section

Data availability

8) Regarding data quantification and statistics, please make sure that the number "n" for how many independent experiments were performed, their nature (biological versus technical replicates), the bars and error bars (e.g. SEM, SD) and the test used to calculate p-values is indicated in the respective figure legends (also for EV figures and all those in an Appendix). Please also check that all the p-values are explained in the legend, and that these fit to those shown in the figure. Please provide statistical testing where applicable. Please avoid the phrase 'independent experiment', but clearly state if these were biological or technical replicates. Please also indicate (e.g. with n.s.) if testing was performed, but the differences are not significant. In case n=2, please show the data as separate datapoints without error bars and statistics. See also: <http://www.embopress.org/page/journal/14693178/authorguide#statisticalanalysis>

9) Please also note our reference format:

10) We updated our journal's competing interests policy in January 2022 and request authors to consider both actual and perceived competing interests. Please review the policy <https://www.embopress.org/competing-interests> and update your competing interests if necessary. Please name this section 'Disclosure and Competing Interests Statement' and put it after the Acknowledgements section.

11) We now use CRediT to specify the contributions of each author in the journal submission system. CRediT replaces the author contribution section. Please use the free text box to provide more detailed descriptions and do NOT provide your final manuscript text file with an author contributions section. See also our guide to authors: <https://www.embopress.org/page/journal/14693178/authorguide#authorshipguidelines>

12) All Materials and Methods need to be described in the main text using our 'Structured Methods' format, which is required for all research articles. According to this format, the Materials and Methods section should include a Reagents and Tools Table (listing key reagents, experimental models, software, and relevant equipment and including their sources and relevant identifiers), uploaded as separate file, followed by a Methods and Protocols section in which we encourage the authors to describe their methods using a step-by-step protocol format with bullet points, to facilitate the adoption of the methodologies across labs. More information on how to adhere to this format as well as downloadable templates (.doc) for the Reagents and Tools Table can be found in our author guidelines (section 'Structured Methods'):

Please add 5 keywords to the manuscript text and order the manuscript sections like this, using these names:
Title page - Abstract - Keywords - Introduction - Results - Discussion - Methods - Data availability section - Acknowledgements - Disclosure and Competing Interests Statement - References - Figure legends - Expanded View Figure legends

I look forward to seeing a revised version of your manuscript when it is ready. Please let me know if you have questions or comments regarding the revision.

Yours sincerely,

Referee #1:

In this manuscript, Misel-Wuchter and colleagues report sophisticated studies in acute LCMV mouse models to determine the necessity of the Tet1/3 DNA demethylases in CD8 T cell responses. Differing from previous findings in Tet2-deficient systems, the data show that loss of Tet1/3 during thymic T cell development results in few baseline differences but does impart a preferential differentiation of CD8 T cells into short-lived effector cells and effector-memory cells following acute LCMV infection. Genome-wide chromatin accessibility and DNA methylation profiling revealed that Tet1/3 poise (license) chromatin regions important for TCR activation during thymic development that are needed for normal responses to acute viral infection. Interestingly, using a CD8-specific inducible Cre driver, the authors determined that Tet1/2 are dispensable for normal effector and memory cell fates following LCMV infection.

The manuscript is well written and logical. The conclusions are strictly drawn from the presented data. The scope of the paper (single model but with multiple causal genetic experiments) fits in the scope of EMBO Reports. My comments are few and relatively minor:

- 1) The text states that "EM-Seq profiling covered an average of 3.6×10^7 individual CpGs with a minimum of 3X coverage of the mouse genome." However, this number exceeds the total number of CpGs usually understood to exist in the mouse genome ($\sim 2 \times 10^7$). Please double check this number and whether the pipeline double-counted palindromic CpGs (i.e., both DNA strands).
- 2) While out of scope for this paper, I am curious whether vitamin C - a non-specific TET activator - rescues any of the effects observed in Tet1/3-deficient systems (for example, see PMID: 34288360 also published in EMBO Reports). The authors have a unique opportunity to determine which, if any, of the effects of vitamin C on T cell activation responses are dependent on Tet1/3. Since the manuscript is excellent as it stands, I will defer to the authors and editor whether to address this question experimentally.
- 3) Can the authors speculate in the Discussion as to whether Tet1 versus Tet3 have distinct effects?

Referee #2:

The study by Misel-Wuchter and colleagues examines the role of the DNA demethylases Tet1/Tet3 in virus-specific CD8+ T cell responses. The present data that shows that CD8+ T cell Tet1/3 deficiency promotes SLEC (effector) differentiation at the cost of MPEC generation. Moreover, at an early memory time point, there are fewer memory CD8+ T cells in both the TEM and TCM compartments. Interestingly, deletion of Tet1/3 in mature CD8+ T cells failed to recapitulate the original observation. This indicates that there is epigenetic programming during thymic development that impacts mature T cell responsiveness.

Overall, this is an interesting paper, and the concept of epigenetic imprinting of functionality during thymic development is

something that is not totally appreciated. Hence, this would be of interest to the field. The major issue with this study is that it falls short in some key areas meaning that more questions are raised than are answered. Certainly if addressed, then this would significantly raise the impact of this study for the field.

1. The major conclusion of this study is that Tet1/3 deficiency promotes SLEC formation at the cost of MPEC generation. The differences observed at the acute (day 8 blood) time point are not particularly impressive. What is more significant is the difference in memory CD8⁺ T cell numbers at the early memory time point (day 28). Given the lack of intrinsic IL-2 production, this would align with earlier studies demonstrating a role for autocrine IL-2 in memory CD8⁺ T cell generation (Williams et al, Nature (2006) 441:890-3. 10.1038/nature04790; Feau et al., Nat Immunol. (2011) 12:908-13. 10.1038/ni.2079). This could be tested by administration of IL-2 (complexed with IL2R) after infection to determine if this is indeed the case.
2. The recall capacity of the resulting memory CD8⁺ T cells after primary infection should also be tested. It would be important to know whether memory potential is in anyway compromised, or is it just the effector expansion that is impacted.
3. The observation that a CD8⁺ T cell intrinsic defect in Tet1/3 was responsible was limited only to day 8. Again, given that CD4⁺ T cell function is important for memory generation, it would have been appropriate to examine memory #s and TCM/TEM subset formation at a memory time point using the P14 Tet1/3 KO cells. Extended the earlier comment, the recall capacity of Tet1/3 KO memory CD8⁺ T cells would also be key to look at.
4. Mechanistically, the impact of Tet1/2 deficiency is not overly apparent. While DMRs are identified, the difference between the WT and KO cells is not overly impressive. What is lacking is a more extensive analysis of non-coding regulatory elements that maybe up and down stream of the gene body. For example, SATB1 has a superenhancer some 300Kb upstream of the TSS. It would be interesting to interrogate whether these regulatory elements (inferred by ATAC-seq data) are lacking demethylation.
5. Is there a correlation of decreased transcription of gene loci that have been identified to have increased methylation status in the Tet1/3 deficient CD8⁺ T cells?
6. As mentioned, the observation that Tet1/3 demethylation is most likely occurring during T cell development is an interesting and important one. The initial epigenetic programming and predetermination of T cell functionality during development is under appreciated, but also an emerging area. However, in this case, it is not confirmed. Given the identification of regions with increased methylation in mature Tet1/3 KO CD8⁺ T cells, determination of when demethylation of these regions occurs during T cell development would go long way to validate and confirm this conclusion. Analysis of DNA methylation status of identified regions in developing T cells (DP stage, SP stage) from WT and Tet1/3 (Rorc-Cre) mice would be adequate.

Referee #3:

In their manuscript, "Tet1/3 regulate CD8 effector and memory T cell fates by licensing the chromatin landscape downstream of T cell receptor activation during thymic development" by Misel-Wuchter et al, the authors use two different models of temporal genetic deletion of Tet1 and Tet3 genes to determine the combined requirement of Tet 1 and Tet3 in supporting CD8 T cell differentiation. The authors find that when Tet1/3 are deleted during the DP stage of thymocyte development, CD8 T cells preferentially differentiate into SLEC after viral infection in a CD8 T cell-intrinsic manner. Interestingly, this is distinctly different from Tet2 deficient CD8 T cells that preferentially differentiate into MPEC. However, deletion of Tet1/3 in mature CD8 T cells has no impact on CD8 T cell differentiation. Taken together, these data identify a role for Tet1/3 in establishing (during thymocyte development) the naïve chromatin state of CD8 T cells that influences T cell differentiation. Overall, the findings are interesting and important for the field. However, there are some key points that should be addressed.

Major Points to Address:

1. What is the individual contribution of Tet1 and Tet3 to the data presented here?
2. The data on SLEC versus MPEC differentiation are quite convincing. The authors follow these data with a series of experiments (Fig. 3) stated to address the mechanism of why Tet1/3 cells skew to SLECs. To this end, they assess IL-2 production, a defect already reported in CD4⁺ T cells by this group. They demonstrate that IL-2 is reduced, but do not address whether reduced IL-2 production impacts the SLEC skewing observed in Tet1/3 mice. In fact, in the discussion, they argue that reduced IL-2 production would not be expected to underlie increased SLEC differentiation. Despite this, they begin to investigate why IL-2 production is reduced. Of the many TCR-derived signals that are required for optimal IL-2 production, it is unclear why only pErk was investigated, and the data ultimately shows that pErk is not affected by Tet1/3 deletion. The authors then jump to assessing Nur77. The rationale for this line of investigation in the context of IL-2 production is unclear, and the finding that Nur77 is reduced in Tet1/3 cDKO mice is again not surprising given their prior publication demonstrating the same phenotype in signaled CD4⁺ thymocytes. Unless these data can support the primary finding of SLEC v. MPEC differentiation, it would be best to move them to a supplemental figure.

Following these data, the authors suggest that Tet1/2 cDKO have decreased TCR signal strength. Lower Nur77 is the only data

to support this statement. Do these cells also have lower CD5 expression or TCR expression?

3. In the presentation of data found in Figure 4, the authors assume that Tet1/3 mediate the observed effects by altering DNA methylation. While this may be case, there are Tet dependent changes that are independent of the protein's enzymatic activity. Moreover, in this part of the manuscript, DNA methylation data has not been presented. Even after the presentation of differential DNA methylation, the changes in methylated regions and gene expression, although convincing, is still only correlative and conclusions should be crafted with this in mind.

4. The authors are commended for the use of E8iCre mice in this study, as the phenotypic differences resulting from the different Cre drivers is striking. They conclude that "Tet1/3 are critical for the epigenetic programming of TCR-responsive regulatory elements in CD8 T cells during thymic development." Presumably, this reprogramming doesn't occur when Tet1/3 is deleted later. It would be helpful to determine the expression of the genes highlighted in Figure E5V in E8iCre Tet1/3 cDKO mice. If reduced levels of these genes (*Itk*, *CD3e*, *Fyn*, *Tcf7*, *Left1*) underlie the phenotypes observed in *Rorc* cDKO mice, one might predict that these genes would not be altered in CD8 T cells using E8iCre to mediate deletion of Tet1/3.

Minor Points:

1. Please indicate the band that is pErk; there are 2 that are induced, one very prominent and one higher and less prominent.
2. Figure 4i - The usefulness of combining the 2 FDR cutoffs in this plot is limited. Is the only red point the one obscured by the legend?
3. For Figure 3D, the bar graph indicates data were gated on CD8 and CD25, but the flow plots don't have a gating strategy. Are the flow plots also pre-gated on CD25+ cells? Are all of the IL-2+ cells CD25+? Is the IL-2 MFI all CD8+ cells, or CD8+ CD25+ cells? Or is it the MFI of just the IL-2 positive cells? (The most informative would be MFI of the IL-2+ cells, otherwise it is simply most reflective of the percent positive. Same question for all MFIs shown.)
4. Please show flow plots for all cytokines.

Referee #1:

In this manuscript, Misel-Wuchter and colleagues report sophisticated studies in acute LCMV mouse models to determine the necessity of the Tet1/3 DNA demethylases in CD8 T cell responses. Differing from previous findings in Tet2-deficient systems, the data show that loss of Tet1/3 during thymic T cell development results in few baseline differences but does impart a preferential differentiation of CD8 T cells into short-lived effector cells and effector-memory cells following acute LCMV infection. Genome-wide chromatin accessibility and DNA methylation profiling revealed that Tet1/3 poise (license) chromatin regions important for TCR activation during thymic development that are needed for normal responses to acute viral infection. Interestingly, using a CD8-specific inducible Cre driver, the authors determined that Tet1/2 are dispensable for normal effector and memory cell fates following LCMV infection.

The manuscript is well written and logical. The conclusions are strictly drawn from the presented data. The scope of the paper (single model but with multiple causal genetic experiments) fits in the scope of EMBO Reports. My comments are few and relatively minor:

We thank the reviewer for their time and valuable feedback on our work.

1) The text states that "EM-Seq profiling covered an average of 3.6×10^7 individual CpGs with a minimum of 3X coverage of the mouse genome." However, this number exceeds the total number of CpGs usually understood to exist in the mouse genome ($\sim 2 \times 10^7$). Please double check this number and whether the pipeline double-counted palindromic CpGs (i.e., both DNA strands).

We sincerely thank the reviewer for pointing out this discrepancy and we apologize. The original text incorrectly stated that "EM-Seq profiling covered an average of 3.6×10^7 individual CpGs," which indeed exceeds the expected number of CpGs in the mouse genome ($\sim 2 \times 10^7$). The correct description is that the analysis retained CpG sites with at least 3X coverage, and the actual average number of individual CpGs covered was 3.6×10^6 , not 3.6×10^7 . We have updated the text to reflect this correction.

2) While out of scope for this paper, I am curious whether vitamin C - a non-specific TET activator - rescues any of the effects observed in Tet1/3-deficient systems (for example, see PMID: 34288360 also published in EMBO Reports). The authors have a unique opportunity to determine which, if any, of the effects of vitamin C on T cell activation responses are dependent on Tet1/3. Since the manuscript is excellent as it stands, I will defer to the authors and editor whether to address this question experimentally.

We thank the reviewer for this thoughtful suggestion. We shared the same initial curiosity following our studies. However, as we delved further into the effects of vitamin C on T cell effector function, we found that the results are not straightforward. Vitamin C acts as a cofactor for Fe(II) and 2-oxoglutarate-dependent dioxygenases, which include TET family enzymes, but also other epigenetic modifiers, such as histone demethylases with Jumonji C (JmjC) domains (PMCID: PMC4506708). In our experiments, we examined the degranulation potential of WT and E8iCre Tet1/3 floxed CD8 T cells following ascorbate treatment (50uM). Interestingly, we observed an increase in degranulation potential (via LAMP1 expression) upon TCR activation, independently of Tet1/3. This information is presented below for the reviewer. However, we

hesitate to attribute this effect solely to vitamin C's influence on Tet2 without further genetic confirmation of Tet2's role. For this reason, we chose not to include this observation in the manuscript. We hope the reviewer agrees with our position that, without definitive genetic evidence confirming Tet2's involvement, this data is incomplete.

Figure for referees not shown.

3) Can the authors speculate in the Discussion as to whether Tet1 versus Tet3 have distinct effects?

We previously reported that Tet1 and Tet3 regulate distinct gene functions in a non-redundant manner in some contexts, but they also share overlapping gene targets. For instance, we found that the impairment of IL-2 was dependent on Tet3, but not Tet1, in CD4 T cells (PMCID: PMC10157375). We regenerated Tet3 single knockouts (which we no longer maintained in our colony) and have found similar observations in CD8 T cells and have included this data in the revised manuscript. However, regardless of its impact on IL-2 production, Tet3 deficiency did not result in increased SLEC/T_{EM} T cell differences, suggesting that this cellular differentiation program likely requires both Tet1 and Tet3 gene targets and is independent of IL-2. These data are now presented in Fig EV3 E-I of the paper, and the roles of Tet1/3 are discussed in greater detail. In future studies, we will be examining what gene targets are uniquely regulated in a Tet1 or Tet3-dependent manner using whole-genome sequencing approaches.

Referee #2:

The study by Misel-Wuchter and colleagues examines the role of the DNA demethylases Tet1/Tet3 in virus-specific CD8⁺ T cell responses. The present data that shows that CD8⁺ T cell Tet1/3 deficiency promotes SLEC (effector) differentiation at the cost of MPEC generation. Moreover, at an early memory time point, there are fewer memory CD8⁺ T cells in both the TEM and TCM compartments. Interestingly, deletion of Tet1/3 in mature CD8⁺ T cells failed to recapitulate the original observation. This indicates that there is epigenetic programming during thymic development that impacts mature T cell responsiveness.

Overall, this is an interesting paper, and the concept of epigenetic imprinting of functionality during thymic development is something that is not totally appreciated. Hence, this would be of interest to the field. The major issue with this study is that it falls short in some key areas meaning that more questions are raised than are answered. Certainly if addressed, then this would significantly raise the impact of this study for the field.

1. The major conclusion of this study is that Tet1/3 deficiency promotes SLEC formation at the cost of MPEC generation. The differences observed at the acute (day 8 blood) time point are not particularly impressive. What is more significant is the difference in memory CD8⁺ T cell numbers at the early memory time point (day 28). Given the lack of intrinsic IL-2 production, this would align with earlier studies demonstrating a role for autocrine IL-2 in memory CD8⁺ T cell generation (Williams et al, Nature (2006) 441:890-3. 10.1038/nature04790; Feau et al., Nat Immunol. (2011) 12:908-13. 10.1038/ni.2079). This could be tested by administration of IL-2 (complexed with IL2R) after infection to determine if this is indeed the case.

We thank the reviewer for this suggestion. We contemplated this experiment. However, since we didn't see 'rescue' of the SLEC and T_{EM} phenotypes in a co-transfer P14 system, where we co-transferred Tet1/3^{CDKO} P14 and WT P14 cells in the same WT host prior to LCMV infection (Fig.2), the data indicated to us that paracrine IL-2 (from WT P14 and host cells) was not sufficient to prevent Tet1/3^{CDKO} skewing into SLECs and T_{EM}S. Administration of IL-2 (complexed with IL2R) after infection would also represent a paracrine scenario. Using distinct models of germline and conditional IL-2 ablation in post-thymic CD8 T cells, a recent study by the Sarkar and Kalia groups (PMID: 35417685), has shown that paracrine IL-2 drives optimal primary expansion, effector and memory differentiation, and metabolic function. In contrast, primary expansion and effector CTL differentiation occur independently of autocrine IL-2 but is uniquely required during primary expansion to program robust secondary expansion potential in memory-fated cells (in agreement with previous studies by others). As such, the defect seen in autocrine IL-2 in Tet1/3^{CDKO} cells using the P14 transfer system do not align with these findings. Furthermore, the IL-2 defect in Tet1/3 deficient mice is Tet3-dependent. Yet, Tet3cDKO mice do not exhibit a skewed SLEC/T_{EM} phenotype (Fig. EV3 G-I), suggesting that additional transcriptional circuits likely drive this process in Tet1/3-deficient mice. This possibility is supported by significant defects in chromatin accessibility in Tet1/3-deficient cells that occur within 18hrs of TCR activation. In future studies, we will be examining what gene targets are uniquely regulated in a Tet1 or Tet3-dependent manner using whole-genome sequencing approaches to drive T_{EM} differentiation.

2. The recall capacity of the resulting memory CD8⁺ T cells after primary infection should also be tested. It would be important to know whether memory potential is in anyway compromised, or is it just the effector expansion that is impacted.

We appreciate the reviewer's suggestion to examine the recall capacity of memory CD8⁺ T cells after primary infection. We agree that it would be important to understand whether memory potential is compromised or whether it is specifically effector expansion that is impacted. However, because the role of thymic pre-programming by Tet enzymes is not well-known at present, our goal was to first establish this paradigm during a primary immune response, which, to our knowledge, has not been demonstrated previously. For this reason, we focused on cell-intrinsic gene functions that are important for CD8 T cell differentiation. As a next step, we will be examining the recall capacity of memory T cells in the absence of thymic epigenetic programming- both CD8 cell-intrinsic mechanisms and CD4-dependent mechanisms. However, this work would require longer experiments and timeframes that were not feasible within the

scope of this revision. These experiments are also challenging given the dramatic reduction of GP-33+ cells at memory time points in Rorc(t)Cre Tet1/3 floxed mice and will require the use of the P14 transfer system.

3. The observation that a CD8+ T cell intrinsic defect in Tet1/3 was responsible was limited only to day 8. Again, given that CD4+ T cell function is important for memory generation, it would have been appropriate to examine memory #s and TCM/TEM subset formation at a memory time point using the P14 Tet1/3 KO cells. Extended the earlier comment, the recall capacity of Tet1/3 KO memory CD8+ T cells would also be key to look at.

We thank the reviewer for this suggestion, and we have examined memory T_{CM}/T_{EM} subset formation at day27 using the P14 co-transfer system. This data is now included in the manuscript as Fig.2E-F. Similar to the earlier time points, we found an increased frequency of T_{EM} cells among Tet1/3 cDKO cells but no difference in the absolute number of T_{EM} cells due to the decrease in total Tet1/3 cDKO recovered at this time point (Fig.2H). In our follow-up manuscript, we will certainly be examining the recall capabilities of these cells alongside the recall capabilities of E8iCre Tet1/3 floxed memory CD8 T cells.

4. Mechanistically, the impact of Tet1/2 deficiency is not overly apparent. While DMRs are identified, the difference between the WT and KO cells is not overly impressive. What is lacking is a more extensive analysis of non-coding regulatory elements that maybe up and down stream of the gene body. For example, SATB1 has a superenhancer some 300Kb upstream of the TSS. It would be interesting to interrogate whether these regulatory elements (inferred by ATAC-seq data) are lacking demethylation.

We acknowledge the point raised by the reviewer regarding distal regulatory elements that may also be affected in Tet1/3 deficient T cells due to the presence of DNA methylation. However, in this work, we did not overlay DMRs over regulatory elements (chromatin-accessible regions) because we defined our methylation probes to be regions within the gene body of a gene+ 3kb upstream. This precludes the ability to statistically examine whether intergenic or distal elements have an increase in DNA methylation in overlapping regions. Furthermore, DMRs tend to be bigger compared to chromatin accessibility regions, can include multiple accessibility sites, and several studies, including a previous study from my lab, have previously demonstrated that DNA methylation in gene body regions can have impacts on distal regulatory elements due to transcriptional regulation that occurs in the contexts of chromatin compartments (PMCID: [PMC6125341](https://pubmed.ncbi.nlm.nih.gov/3125341/)). Lastly, intergenic regions are difficult to annotate as they may be more proximal to other genes. To address the reviewer's comments, we have instead performed a detailed analysis to examine whether there is a correlation between reduced chromatin accessibility in distinct regulatory regions (promoter, intragenic and intergenic) that are annotated to gene loci in which DNA methylation differences exist in cis (within the gene body+3kb upstream of the promoter) between activated WT and Tet1/3cDKO T cells. This analysis revealed that a large subset of gene loci in which methylation is increased in Tet1/3cDKO CD8 T cells have reduced chromatin accessibility in promoter, intergenic, and intragenic regions, while only a handful show such a correlation when methylation is reduced. This analysis is presented in Fig. 5J. From this analysis, we found that SATB1 also abides by this rule and this data is included here for the reviewer. Notably, there are also sites of increased

methylation in the intergenic space in which accessibility is reduced and DNA methylation is higher in Tet1/3cDKO cells.

5. Is there a correlation of decreased transcription of gene loci that have been identified to have increased methylation status in the Tet1/3 deficient CD8⁺ T cells?

Yes, as was previously shown in Fig. EV5C, the mRNA expression of *Itk*, *Cd3e*, *Fyn* and *Tcf7* in Tet1/3 cDKO CD8 T cells were reduced compared to WT controls. We also have protein expression data showing that CD3e protein levels are reduced in CD8 T cells from RorcCre Tet1/3 floxed but not E8iCre Tet1/3 floxed mice (Fig. EV5D, Fig. EV6L), in concordance with increased methylation and reduced expression of the *Cd3e* gene.

6. As mentioned, the observation that Tet1/3 demethylation is most likely occurring during T cell development is an interesting and important one. The initial epigenetic programming and predetermination of T cell functionality during development is under appreciated, but also an emerging area. However, in this case, it is not confirmed. Given the identification of regions with increased methylation in mature Tet1/3 KO CD8⁺ T cells, determination of when demethylation of these regions occurs during T cell development would go long way to validate and confirm this conclusion. Analysis of DNA methylation status of identified regions in developing T cells (DP stage, SP stage) from WT and Tet1/3 (Rorc-Cre) mice would be adequate.

We thank the reviewer for recognizing the paucity of appreciation for the importance of epigenetic predetermination in T cell functionality during development. Regarding the timing of DNA demethylation during thymic development, we have previously shown that this process occurs during lineage specification of CD4 and CD8 cells from DP precursors, with **Tet1/3** playing a particularly important role—specifically in mediating DNA demethylation across the CD4 gene locus in thymocytes (PMCID: PMC6125341). These findings were obtained using Rorc(t)^{Cre} Tet1/3 floxed mice and thus provide clear evidence of the active role of Tet1/3 during thymic development. Furthermore, these findings align with those of another study showing that 5hmC levels increase dramatically in CD4⁺ and CD8⁺ SP cells following their differentiation from DP cells (PMCID: PMC4136618)

We have performed global quantifications of 5hmC levels in pre-selected DP (CD69-CD24^{hi}CD4⁺CD8⁺) and post-selected CD8 SP thymocytes by ELISA and this data is now included in Fig. 6G. Consistent with previous reports, our data show that 5hmC marks are more abundant in post-selected CD8 SP thymocytes compared to pre-selected DPs. However, 5hmC levels are dramatically reduced in CD8 SP cells isolated from Rorc(t)^{Cre} Tet1/3 floxed mice but

not in CD8 SP isolated from E8i^{Cre} Tet1/3 floxed mice. While this analysis does not examine specific loci, we hope it still conveys the point that Tet1/3-mediated DNA demethylation is crucial during thymic development and taken together with other evidence presented (e.g RNA and protein expression of genes) suggests that this programming is essential for T cell function.

Referee #3:

In their manuscript, "Tet1/3 regulate CD8 effector and memory T cell fates by licensing the chromatin landscape downstream of T cell receptor activation during thymic development" by Misel-Wuchter et al, the authors use two different models of temporal genetic deletion of Tet1 and Tet3 genes to determine the combined requirement of Tet 1 and Tet3 in supporting CD8 T cell differentiation. The authors find that when Tet1/3 are deleted during the DP stage of thymocyte development, CD8 T cells preferentially differentiate into SLEC after viral infection in a CD8 T cell-intrinsic manner. Interestingly, this is distinctly different from Tet2 deficient CD8 T cells that preferentially differentiate into MPEC. However, deletion of Tet1/3 in mature CD8 T cells has no impact on CD8 T cell differentiation. Taken together, these data identify a role for Tet1/3 in establishing (during thymocyte development) the naïve chromatin state of CD8 T cells that influences T cell differentiation. Overall, the findings are interesting and important for the field. However, there are some key points that should be addressed.

We thank the reviewer for their feedback and helpful comments on our work.

Major Points to Address:

1. What is the individual contribution of Tet1 and Tet3 to the data presented here?

We previously reported that Tet1 and Tet3 regulate distinct gene functions in a non-redundant manner in some contexts, but they also share overlapping gene targets. For instance, we found that the impairment of IL-2 was dependent on Tet3, but not Tet1, in CD4 T cells (PMCID: PMC10157375). We regenerated Tet3 single knockouts (which we no longer maintained in our colony) and have found similar findings in CD8 T cells and have included this data in the revised manuscript in Fig. EV3E. However, despite the profound impact on IL-2 production, Tet3 deficiency did not result in increased SLEC T cell differences (Fig. EV3G-I), suggesting that this cellular differentiation program likely requires both Tet1 and Tet3 gene targets and is not strictly driven by IL-2. We expand briefly on the individual roles of Tet1/3 in the revised manuscript. In future studies, we will be examining what gene targets are uniquely regulated in a Tet1 or Tet3-dependent manner using whole-genome sequencing approaches.

2. The data on SLEC versus MPEC differentiation are quite convincing. The authors follow these data with a series of experiments (Fig. 3) stated to address the mechanism of why Tet1/3 cells skew to SLECs. To this end, they assess IL-2 production, a defect already reported in CD4+ T cells by this group. They demonstrate that IL-2 is reduced, but do not address whether reduced IL-2 production impacts the SLEC skewing observed in Tet1/3 mice. In fact, in the discussion, they argue that reduced IL-2 production would not be expected to underlie increased SLEC differentiation. Despite this, they begin to investigate why IL-2 production is reduced. Of the many TCR-derived signals that are required for optimal IL-2 production, it is unclear why only

pErk was investigated, and the data ultimately shows that pErk is not affected by Tet1/3 deletion. The authors then jump to assessing Nur77. The rationale for this line of investigation in the context of IL-2 production is unclear, and the finding that Nur77 is reduced in Tet1/3 cDKO mice is again not surprising given their prior publication demonstrating the same phenotype in signaled CD4⁺ thymocytes. Unless these data can support the primary finding of SLEC v. MPEC differentiation, it would be best to move them to a supplemental figure.

We apologize for the lack of clarity behind our thought process and rationale in Fig. 3. Our intent was not to draw an emphasis on the IL-2 deficiency but to highlight how Tet1/3 affects distinct epigenetic and transcriptomic nodes downstream of TCR activation. In response to TCR stimulation, Tet1/3 cDKO cells showed a decrease in TCR signaling strength (Nur77 and CD5- Fig. 3I-J) and a decrease in IL-2 production, which was not rescued by increasing the strength of TCR signaling (by providing different amounts of TCR ligands/stimuli, Fig. 3K). As we discussed in point 1 above, IL-2 deficiency is not sufficient to skew SLEC/T_{EM} phenotypes as Tet3 cKO cells do not show similar skewing. Furthermore, a recent study by the Sarkar and Kalia groups (PMID: 35417685), has shown that primary effector CTL differentiation occurs independently of autocrine IL-2 but is uniquely required during primary expansion to program robust secondary expansion potential in memory-fated cells (in agreement with previous studies by others). Although the evidence is indirect, our interpretation is that IL-2 is likely not the primary reason why Tet1/3 deficiency leads to SLEC/T_{EM} skewing but that other TCR-dependent signals and Tet1/3-dependent transcriptional circuits drive this phenotype.

Upon pMHC engagement, the TCR triggers the activation of Erk and NFAT pathways. These pathways are rapidly activated within minutes of antigen encounter, and over these short timescales, they appear to function in an all-or-nothing, 'digital' manner regardless of pMHC affinity or dose. (PMCID: PMC1262625, PMCID: PMC8536361 , PMCID: PMC6598675). Erk signaling leads to nuclear accumulation and activation of the AP-1 family TFs (PMCID: PMC5632840). As we could not rescue the IL-2 phenotype in Tet1/3cDKO cells by increasing TCR signaling strength, we examined whether ERK activation pathways were defective in these cells, but we did not see a difference. Moreover, our focus was on ERK as a large fraction of chromatin regions where AP-1 factors bind (e.g. JunB) were most affected in Tet1/3 cDKO cells. Notably, IL-2 production is also dependent on JunB and ERK pathway activation. We have explained this rationale more carefully and revised our Fig.3 title to “Tet1 and Tet3 are both required to restrict SLEC/T_{EM} CD8 T cell differentiation by controlling TCR-dependent epigenetic circuits”. We hope to have addressed the reviewer’s point and thank the reviewer for pointing out the lack of clarity.

Following these data, the authors suggest that Tet1/2 cDKO have decreased TCR signal strength. Lower Nur77 is the only data to support this statement. Do these cells also have lower CD5 expression or TCR expression?

Yes indeed, Tet1/3 cDKO T cells have lower protein expression of CD3e, CD5 and TCRβ expression as measured by flow cytometry, in support of the conclusion that TCR signal strength is reduced in Tet1/3 cDKO T cells, by proxy of markers commonly used to assess TCR signal strength. This data is now included in Fig.3K, Fig.EV5D, Fig.EV6K-L.

3. In the presentation of data found in Figure 4, the authors assume that Tet1/3 mediate the observed affects by altering DNA methylation. While this may be case, there are Tet dependent

changes that are independent of the protein's enzymatic activity. Moreover, in this part of the manuscript, DNA methylation data has not been presented. Even after the presentation of differential DNA methylation, the changes in methylated regions and gene expression, although convincing, is still only correlative and conclusions should be crafted with this in mind.

We thank the reviewer for pointing this out and we acknowledge that the catalytic independent functions of Tet1/3 may play a role in controlling differentiation programs in CD8 T cells. We have referenced the appropriate literature and have modified our conclusions to include this possibility.

4. The authors are commended for the use of E8iCre mice in this study, as the phenotypic differences resulting from the different Cre drivers is striking. They conclude that "Tet1/3 are critical for the epigenetic programming of TCR-responsive regulatory elements in CD8 T cells during thymic development." Presumably, this reprogramming doesn't occur when Tet1/3 is deleted later. It would be helpful to determine the expression of the genes highlighted in Figure E5V in E8iCre Tet1/3 cDKO mice. If reduced levels of these genes (Itk, CD3e, Fyn, Tcf7, Left1) underlie the phenotypes observed in Rorc cDKO mice, one might predict that these genes would not be altered in CD8 T cells using E8iCre to mediate deletion of Tet1/3.

We thank the reviewer for the excellent suggestion. We have examined the expression of Itk, CD3e, Fyn, Tcf7, and Lef1 in E8iCre Tet1/3 floxed CD8 T cells. Our findings show that while expression of these genes is reduced in Rorc(t)^{Cre} Tet1/3 floxed T cells, they remain unaffected in E8i^{Cre} Tet1/3 floxed T cells, providing further support for the critical role of Tet1/3 in the early programming of these genes. We further show that the protein levels of CD3e is reduced in Rorc(t)^{Cre} Tet1/3 floxed but not in E8i^{Cre} Tet1/3 floxed CD8 T cells. This data is now shown in Fig. EV6L

Minor Points:

1. Please indicate the band that is pErk; there are 2 that are induced, one very prominent and one higher and less prominent.

We have indicated the identity of pERK1 and pERK2 on the blots.

2. Figure 4i - The usefulness of combining the 2 FDR cutoffs in this plot is limited. Is the only red point the one obscured by the legend?

We apologize for this image formatting issue and have amended the figure to rectify this. Now all points with the indicated FDR cut-offs are visible.

3. For Figure 3D, the bar graph indicates data were gated on CD8 and CD25, but the flow plots don't have a gating strategy. Are the flow plots also pre-gated on CD25+ cells? Are all of the IL-2+ cells CD25+? Is the IL-2 MFI all CD8+ cells, or CD8+ CD25+ cells? Or is it the MFI of just the IL-2 positive cells? (The most informative would be MFI of the IL-2+ cells, otherwise it is simply most reflective of the percent positive. Same question for all MFIs shown.)

We apologize for the lack of clarity. We always pre-gate on activated T cells (CD25+ or CD69+) which represent >90% of the total cells, to ensure examination of cytokines within the activated T cell compartment. This is now indicated on the FACS plots and legends. Notably, upon TCR-activation, we do not observe a difference in the frequency or MFI of CD25+ cells or CD69+ cells (Fig. EV3B) in the absence of Tet1/3. We have examined gMFI of IL-2 among IL2+ cells in Fig. 3D and found that the IL-2 levels among IL2-producing cells are not statistically different

between WT and Tet1/3 cDKO cells, suggesting that the defect lie predominantly in the ability of a proportion of Tet1/3 cells to secrete IL-2 upon activation. Furthermore, we have included gMFI analyses within cytokine+ populations for all experiments.

4. Please show flow plots for all cytokines.

We now ensure that all cytokine flow plots are shown, namely, IFNg-producing cells in Fig. 6D, IFNg-a and TNF-a-producing cells in Fig.2C, 2D

Dear Dr. Issuree

Thank you for the submission of your revised manuscript to our editorial offices. I have now received the reports from the three referees that were asked to re-evaluate the study, you will find below. As you will see, the referees now support the publication of the study in EMBO reports. However, referee #3 has remaining concerns and suggestions to improve the study, I ask you to address in a final revised manuscript. Please also provide a final p-b-p-response addressing the remaining points of the referee.

Moreover, I have these editorial requests I also ask you to address:

- Please provide a more compact title with not more than 100 characters (including spaces).
 - Please add up to 5 keywords to the manuscript and order the sections like this, using these names:
Title page - Abstract - Keywords - Introduction - Results - Discussion - Methods - Data availability section - Acknowledgements (including the funding information) - Disclosure and Competing Interests Statement - References - Figure legends - Expanded View Figure legends
 - Thus, please add the funding information to the Acknowledgements.
 - Please also make sure that all the funding information is entered into the online submission system and that it is complete and similar to the one in the acknowledgement section of the manuscript text file. Presently, grants from the Fibrosis Foundation University of Iowa RDP Bioinformatics Core; F31A1164640; T32AI007511 are missing from the submission system. Please check.
 - We now use CRediT to specify the contributions of each author in the journal submission system. CRediT replaces the author contribution section. Please use the free text box to provide more detailed descriptions and do NOT provide your final manuscript text file with an author contributions section. See also our guide to authors:
<https://www.embopress.org/page/journal/14693178/authorguide#authorshipguidelines>
 - The Data availability section (DAS) is restricted for information on primary datasets produced in a study that are deposited in a public database. Please remove any other information from this section. Please also remove now the referee access information from the Data Availability section and make sure the datasets are public latest upon online publication of the paper.
 - Please use our reference format:
<http://www.embopress.org/page/journal/14693178/authorguide#referencesformat>
 - Please make sure that all figure panels are called out separately and sequentially. Presently, there seems to be no callout for Fig. 2D. Please check.
 - Please check again that the number "n" for how many independent experiments were performed, their nature (biological versus technical replicates), the bars and error bars (e.g. SEM, SD) and the test used to calculate p-values is indicated in the respective figure legends. Please also check that all the p-values are explained in the legend, and that these fit to those shown in the figure. Please provide statistical testing where applicable. Please avoid the phrase 'independent experiment', but clearly state if these were biological or technical replicates. Please also indicate (e.g. with n.s.) if testing was performed, but the differences are not significant. In case n=2, please show the data as separate datapoints without error bars and statistics. See also:
<http://www.embopress.org/page/journal/14693178/authorguide#statisticalanalysis>
- If n<5, please show single datapoints for diagrams. Moreover:
- Please define the annotated p values ****/***/**/* as well as provide the exact p-values for the same in the legend of figure EV5 C, D as appropriate.
 - Please note that the exact p values are not provided in the legends of figures 1A-E; 2B-F; 3C, D, G, H, I, J, K; 6G, EV1 D, E, G, H; EV2 A, D, E; EV3 A, E; EV6 K, L.
 - Please indicate the statistical test used for data analysis in the legends of figures 5F, EV5 A.
 - Please note that in figures 1A-E; 2B-F; 3C, D, G, H, I, J, K; 6G, EV1 D, E, G, H; EV2 A, D, E; EV3 A, E; EV6 K, L there is a mismatch between the annotated p values in the figure legend and the annotated p values in the figure file that should be corrected.
 - Please modify the legends of the five datasets. The items should have the same name ('Dataset EVx') in the legend, not 'Table x').
 - Please remove Figure 7 and upload this as synopsis image without the text. The text can be used for the synopsis text and/or the bullet points (see below).

- Please remove the table from the manuscript text file. Please upload the table separately as 'Reagents and Tools Table' and add callouts to the table in the Methods section where appropriate. More information on how to adhere to this format as well as downloadable templates (.doc) for the Reagents and Tools Table can be found in our author guidelines (section 'Structured Methods'):

- Please remove the sentence 'Source data are available online for this figure' from the legends.

In addition, I would need from you uploaded separately:

Best,

Referee #1:

The authors have addressed my critiques.

Referee #2:

The authors have adequately addressed my concerns. While I think analysis of memory recall responses is of importance, I accept their arguments that they are examining establishment of memory, and that this perhaps might be better suited to a follow up study.

Referee #3:

Summary: The authors have addressed most of the concerns noted in my initial review. However, based on the comments below, the authors need to be more rigorous in their data interpretation and precise in their wording surrounding whether the phenotypes they are describing are dependent on Tet1 and Tet3, as the additional data point to the fact that this could be largely Tet1. This needs to be acknowledged.

Specific points: The authors indicate that they evaluated gene expression of specific genes in the Ei8cre system that had been identified as targets in the Rorc(t)Cre model. They present these data in Fig EV6 J-L. However, these panels are never referred to in the text. Moreover, it is not clear that the data were analyzed the same way as in the corresponding Fig. EV5, or perhaps the WT samples are just much more variable in the new data (EV6.J)? There is also no discussion (that I could see) on the individual role of Tet1 in contributing to these phenotypes and the possibility that all of the phenotype in SLEC v MPEC might be due to loss of Tet1 alone.

Some of the wording has not been adjusted to reflect the findings based on the requested line of investigation. For example, they partially addressed whether the findings were driven predominantly by loss of Tet1 or Tet3. The authors demonstrated that loss of Tet3 alone was responsible for the IL-2 defect observed in Tet1/3 dKO cells, but the differences in SLEC vs MPEC were not driven by Tet3 loss. Therefore, the following headings are not correct:

- "Tet1 and Tet3 restrain the differentiation of SLECs and effector memory T cells in a cell intrinsic manner" Rather, their data show that Tet3 deficiency alone does not contribute to this phenotype, and they did not address the independent role of Tet1. It is formally possible that these findings are solely that of a Tet1-deficiency. Therefore, the correct heading should be "Loss of Tet1 and Tet3 restrains the differentiation...cell intrinsic manner"

- Similarly, "Tet1 and Tet3 are both required to restrict SLEC/TEM CD8 T cell differentiation by controlling TCR-dependent epigenetic circuits" is not an accurate statement for the same reasoning.

In evaluating the Tet3 cKO, the authors perform the same LCMV infection and assess CD127 vs KLRG1 expression in CD8+ gp33+ cells (Fig EV3 panel H). In this figure, they show that CD8 differentiation in the WT and the Tet3cKO cells is very similar. However, the percent of CD8+ cells that are KLRG1+ in the WT is nearly 80%. This frequency is very different than the WT phenotype seen in Figg. 1D where ~45% of the WT gp33+ cells were KLRG1+. The dramatic difference in WT phenotype between these two figures makes it difficult to draw a strong conclusion.

Reviewer#3

The authors have addressed most of the concerns noted in my initial review. However, based on the comments below, the authors need to be more rigorous in their data interpretation and precise in their wording surrounding whether the phenotypes they are describing are dependent on Tet1 and Tet3, as the additional data point to the fact that this could be largely Tet1. This needs to be acknowledged.

We thank the reviewer for their valuable input. We have modified our text in the discussion to clarify that the role of Tet1 in CD8 T cell fates cannot be excluded and that genetic analyses complemented with epigenetic profiling are needed in future studies to fully understand the independent/ co-dependent roles of Tet1 and Tet3 in CD8 T cell differentiation. Furthermore, we have amended our conclusion for Fig 3 - "As autocrine IL-2 is not required for effector CD8 T cell differentiation during primary infection (Toumi *et al*, 2022) and considering the lack of SLEC/T_{EM} phenotypes in Tet3 cKO mice, we conclude that Tet3 is not sufficient to restrict SLEC/T_{EM} CD8 T cell differentiation and that Tet1 or the combined actions of Tet1 and Tet3 are likely required for this process."

The authors indicate that they evaluated gene expression of specific genes in the Ei8cre system that had been identified as targets in the Rorc(t)Cre model. They present these data in Fig EV6 J-L. However, these panels are never referred to in the text. Moreover, it is not clear that the data were analyzed the same way as in the corresponding Fig. EV5, or perhaps the WT samples are just much more variable in the new data (EV6.J)? There is also no discussion (that I could see) on the individual role of Tet1 in contributing to these phenotypes and the possibility that all of the phenotype in SLEC v MPEC might be due to loss of Tet1 alone.

We sincerely apologize for missing a callout in the text and now refer to Fig EV6 J-L in the main text. We have also reanalyzed our data Fig. EV6J to match the analysis previously performed in EV5. The difference is that in the present analysis, each WT mouse is normalized to 1, and the DKO is presented as the fold change to its littermate WT. We previously performed averages of all the WT animals but the data was a combination of 2 independent experiments with 2 littermate mice/group/exp, which is why the data appeared noisy.

Some of the wording has not been adjusted to reflect the findings based on the

requested line of investigation. For example, they partially addressed whether the findings were driven predominantly by loss of Tet1 or Tet3. The authors demonstrated that loss of Tet3 alone was responsible for the IL-2 defect observed in Tet1/3 dKO cells, but the differences in SLEC vs MPEC were not driven by Tet3 loss. Therefore, the following headings are not correct:

- "Tet1 and Tet3 restrain the differentiation of SLECs and effector memory T cells in a cell intrinsic manner" Rather, their data show that Tet3 deficiency alone does not contribute to this phenotype, and they did not address the independent role of Tet1. It is formally possible that these findings are solely that of a Tet1-deficiency. Therefore, the correct heading should be "Loss of Tet1 and Tet3 restrains the differentiation...cell intrinsic manner"
- Similarly, "Tet1 and Tet3 are both required to restrict SLEC/TEM CD8 T cell differentiation by controlling TCR-dependent epigenetic circuits" is not an accurate statement for the same reasoning.

We thank the reviewer for their helpful recommendation. We acknowledge that the wording of our conclusions could be misleading, and we have amended most of the titles to avoid confusion. However, we would like to clarify that, we avoided the use Tet1 and Tet3 individually but rather refer to them as Tet1/3 because we do not distinguish between the individual/co-dependent roles of Tet1 and Tet3. This was necessary in instances where the syntax of a sentence is even more confusing to a reader (e.g. when discussing the loss of something guiding a specific mechanism or promoting/restraining something else) or when the titles would be overly repetitive, because it is always going to the loss of Tet1 and Tet3 that results in something. As such we hope the mention of the "loss of Tet1/3" wherever possible and the use of "Tet1/3" serve as better choices without creating confusion, and address the reviewer's concerns. The amended conclusion titles are:

1. Loss of Tet1/3 in developing CD8 T cells results in increased SLECs and effector memory T cell differentiation during acute LCMV infection
2. Tet1/3 restrain the differentiation of SLECs and effector memory T cells in a cell-intrinsic manner
3. Tet1/3 restrict SLEC/T_{EM} CD8 T cell differentiation by controlling TCR-dependent epigenetic circuits
4. Loss of Tet1/3 impairs chromatin accessibility at regulatory regions in TCR-responsive genes
5. CD8 T cells undergo Tet1/3-dependent DNA demethylation during thymic development

6. Tet1/3 are dispensable in peripheral CD8 T cells for effector and memory differentiation

In evaluating the Tet3 cKO, the authors perform the same LCMV infection and assess CD127 vs KLRG1 expression in CD8+ gp33+ cells (Fig EV3 panel H). In this figure, they show that CD8 differentiation in the WT and the Tet3cKO cells is very similar. However, the percent of CD8+ cells that are KLRG1+ in the WT is nearly 80%. This frequency is very different than the WT phenotype seen in Fig. 1D where ~45% of the WT gp33+ cells were KLRG1+. The dramatic difference in WT phenotype between these two figures makes it difficult to draw a strong conclusion.

We acknowledge that although we now rigorously test our in house generated stocks by measuring viral titers using plaque assays, we have observed that viral stocks made on different times and stored for years and/or shared across labs can lead to variability in infections/SLECs. We have however consistently observed 60-80% SLECs on day 8 dpi with LCMV Armstrong, which is what is shown in Fig. EV3 and Fig. 2C. Despite variable infections with different stocks, we consistently observe higher SLECs in Tet1/3 DKO mice over numerous experiments (independent of the magnitude of infections), and we are confident of our conclusions that Tet1/3 deficiency significantly skews SLEC differentiation, while Tet3 deficiency alone is not sufficient for significant skewing. That said, we acknowledge that a time-course looking at SLEC formation at earlier time-points could provide a better appreciation of the role of Tet3 in this process and acknowledge this in the text. However, given the time constraint and the limited number of mice after the rederivation of Tet3 single KO mice, we could not perform a more granular dissection.

Dr. Priya Issuree
University of Iowa
Internal Medicine
431 Newton Road
EMRB
Iowa City, IA 52242
United States

Dear Dr. Issuree,

I am very pleased to accept your manuscript for publication in the next available issue of EMBO reports. Thank you for your contribution to our journal.

Yours sincerely,
